# Revealing evolution of tropane alkaloid biosynthesis by analyzing two genomes in the Solanaceae family

Fangyuan Zhang[1,2], Fei Qiu[1,2], Junlan Zeng[1,2], Zhichao Xu [3], Yueli Tang[1,2], Tengfei Zhao[1,2], Yuqin Gou[1,2], Fei Su[1,2], Shiyi Wang[1,2], Xiuli Sun[1,2], Zheyong Xue[3], Weixing Wang[4], Chunxian Yang[1,2], Lingjiang Zeng[1,2], Xiaozhong Lan[5], Min Chen[6], Junhui Zhou[7] & Zhihua Liao [1,2] ✉

Tropane alkaloids (TAs) are widely distributed in the Solanaceae, while some important medicinal tropane alkaloids (mTAs), such as hyoscyamine and scopolamine, are restricted to certain species/tribes in this family. Little is known about the genomic basis and evolution of TAs biosynthesis and specialization in the Solanaceae. Here, we present chromosome-level genomes of two representative mTAs-producing species: *Atropa belladonna* and *Datura stramonium*. Our results reveal that the two species employ a conserved biosynthetic pathway to produce mTAs despite being distantly related within the nightshade family. A conserved gene cluster combined with gene duplication underlies the wide distribution of TAs in this family. We also provide evidence that branching genes leading to mTAs likely have evolved in early ancestral Solanaceae species but have been lost in most of the lineages, with *A. belladonna* and *D. stramonium* being exceptions. Furthermore, we identify a cytochrome P450 that modifies hyoscyamine into norhyoscyamine. Our results provide a genomic basis for evolutionary insights into the biosynthesis of TAs in the Solanaceae and will be useful for biotechnological production of mTAs via synthetic biology approaches.

Solanaceae is one of the largest families of Angiosperm plants and includes economically important species such as tomatoes, potatoes, and eggplants. In addition to cultivated species, several species have also attracted considerable interest due to their lethal toxicity and medicinal value, such as *Atropa belladonna* L., *Hyoscyamus niger* L., and *Datura stramonium* L. These plants have been used globally as poisons, hallucinogens, and anesthetic agents for a long time in both the New and Old Worlds, as their usage can be traced back to ancient Egypt in 1500 BC[1]. The main pharmacological substances in those plants are hyoscyamine and scopolamine. They are still utilized to treat neuromuscular disorders such as Parkinson's disease or used as anesthetics and analgesics, and antidotes to nerve agent[2], which are

[1]State Key Laboratory of Silkworm Genome Biology, School of Life Sciences, Southwest University, Chongqing 400715, China. [2]Integrative Science Center of Germplasm Creation in Western China (Chongqing) Science City & Southwest University, SWU-TAAHC Medicinal Plant Joint R&D Centre, Southwest University, Chongqing 400715, China. [3]Heilongjiang Key Laboratory of Plant Bioactive Substance Biosynthesis and Utilization, Northeast Forestry University, Harbin, Heilongjiang 150040, China. [4]College of Horticulture and Landscape Architecture, Southwest University, Chongqing 400715, China. [5]TAAHC-SWU Medicinal Plant Joint R&D Centre, Tibetan Collaborative Innovation Centre of Agricultural and Animal Husbandry Resources, Xizang Agricultural and Animal Husbandry College, Nyingchi, Tibet 860000, China. [6]College of Pharmaceutical Sciences, Key Laboratory of Luminescent and Real-Time Analytical Chemistry (Ministry of Education), Southwest University, Chongqing 400715, China. [7]State Key Laboratory of Dao-di Herbs, National Resource Center for Chinese Materia Medica, China Academy of Chinese Medical Sciences, Beijing 100700, China. ✉e-mail: zhliao@swu.edu.cn

listed in the Model Lists of Essential Medicines by the World Health Organization. Hyoscyamine and scopolamine are tropane alkaloids (TAs), which are widespread in some distantly related plant families, including Solanaceae, Erythroxylaceae, Euphorbiaceae, Rhizoporaceae, and Convolvulaceae[3]. The scattered distribution of TAs along the Eudicots branch is considered as the result of convergent evolution[4].

TAs are characterized by an 8-azabicyclo [3.2.1] octane core skeleton containing a cycloheptane ring with a nitrogen bridge[4]. There is a diverse class of ~300 specialized TAs, which may have evolved in response to strong natural selection[5]. Structurally, almost all TAs are esters of various organic acids, including tropic, benzoic, cinnamic, isovaleric, and tiglic acids conjugated to hydroxylated tropane derivatives[4]. However, not all TAs have defined physiological activity. TAs used as medicinal substances are therefore termed medicinal tropane alkaloids (mTAs), especially anticholinergic hyoscyamine and scopolamine in the Solanaceae.

After more than a century of research, it was not until 2020 that the complete biosynthetic route of hyoscyamine and scopolamine was clarified[4]. Thirteen enzymes are involved in the biosynthesis of scopolamine from two starting amino acid precursors, ornithine and phenylalanine. The complete biosynthetic pathway of mTAs can be divided into three modules (see below): I, biosynthesis of the core structure of TAs, tropine; II, biosynthesis of the tropyl moiety; and III, modification followed by condensation of tropine and tropyl. In the module I, ornithine decarboxylase (ODC[6]), putrescine *N*-methyltransferase (PMT[7]), *N*-methylputrescine oxidase (MPO), type III polyketide synthase (PYKS[8,9]), tropinone synthase (CYP82M3[9]), and tropine reductase (TRI[10]) are involved. In module II, biosynthesis of the tropyl moiety, aromatic amino acid aminotransferase 4 (AT4[11]), phenylpyruvic acid reductase (PPAR[12]), and phenyllactate UDP-glycosyltransferase (UGT1[13]) leads to the production of phenyllactylglucose. Subsequently, in module III, the tropine esterification of phenyllactylglucose is catalyzed by littorine synthase (LS[13]), leading to the production of littorine, which was successively modified by littorine mutase (CYP80F1[14]), hyoscyamine dehydrogenase (HDH[15]), and hyoscyamine 6β-hydroxylase (H6H[13]) to generate scopolamine. Modules I and II were termed upstream pathways; module III was termed the downstream pathway and the mTAs-specific pathway. All the identified enzymes leading to the production of hyoscyamine, and scopolamine are dominantly expressed in the roots, especially the secondary roots, of mTAs-producing plants, such as *A. belladonna* and *D. stramonium*. In addition to scopolamine, norhyoscyamine, resulting from the *N*-demethylation of hyoscyamine, has also been identified in mTAs-producing species in Solanaceae, such as the *Anthrocercis albicans × Duboisia myoporoides* hybrid and *Datura innoxia*[16]. Norhyoscyamine is an important intermediate that produces ipratropium bromide, an anticholinergic drug used in the control of symptoms related to bronchospasm in chronic obstructive pulmonary disease (COPD) and asthma[17]. However, there is still limited information on the enzymes that convert hyoscyamine to norhyoscyamine.

Although the chemical and enzymatic routes of scopolamine production have been identified and used for scopolamine biosynthesis in yeast, the complete biosynthesis route has not been revealed in any species. Different mTAs producing plants were used by different research groups to elucidate each reaction step. In the 1990s, Hashimoto and coworkers identified the enzymes and the corresponding genes of PMT, MPO, TRI, and H6H mainly in *D. stramonium*, *H. niger*, and *Nicotiana tabacum*[18–20]. Subsequently, in the 2000s, a few independent research groups identified the other genes mainly in *A. belladonna* and *D. stramonium*. The most intensively studied species in relation to the biosynthetic pathway of mTAs is *A. belladonna*, in which all the genes participating in scopolamine biosynthesis have been identified except *MPO* and *TRI*. In another well-known mTAs-producing plant, *D. stramonium*, only a few genes involved in mTAs

biosynthesis have been reported, including *PMT*, *TRI*, *HDH*, and *H6H*. *A. belladonna* and *D. stramonium* belong to two main tribes that produce mTAs, Hyoscyameae (including *Atropa*, *Anisodus*, and other genera) and Daturaeae (including *Datura*, *Brugmansia*, and other genera), respectively, which belong to two early-diverging clades[21]. Interestingly, the distribution centers of Atropa and Datura are quite different; Atropa is distributed exclusively in Eurasia, while Datura is mainly distributed in the New World[22]. Thus, whether the same biosynthetic genes are employed by *A. belladonna* and *D. stramonium* to produce the same mTAs remains unclear. Furthermore, how homologous genes among distant species in the Solanaceae family evolved to produce tropine, the chemical foundation of TAs that are widely distributed in this family, has not yet been determined.

In this work, we present two chromosome-level genomes of two medicinal plants (*A. belladonna* and *D. stramonium*) of Solanaceae, which not only provide insights into the diversification of TAs biosynthetic pathway but also provide valuable bioinformatic and genetic resources that could be used to enhance the production of mTAs by genetic engineering of plants or microbes[23,24]. By employing phylogenetic analysis along with synteny analysis, we find that the biosynthetic pathway of TAs is highly conserved in Solanaceae, at least in module I and II. Moreover, we uncover a conserved gene cluster for tropine biosynthesis, composed of *TRI* and *CYP82M3*. Combined with *TRI* expansion driven by tandem duplication and/or whole-genome triplication (WGT), this gene cluster provides a basis for inferring the widespread distribution of TAs in this family. Nevertheless, the loss of the mTAs-specific genes, *LS* and *CYP80F1*, shapes the uneven distribution of mTAs in the Solanaceae. Furthermore, we identify a cytochrome P450 gene that catalyzes the *N*-demethylation of hyoscyamine to generate norhyoscyamine. These findings contribute to our understanding of the evolution of TAs in Solanaceae and will be valuable for metabolic engineering of mTAs.

## Results
### Genome sequencing, assembly, and annotation of two species

Using *K*-mer analysis with short reads (Supplementary Fig. 1), the genome size of *A. belladonna* was estimated to be ~1.65 Gb and that of *D. stramonium* was estimated to be ~1.80 Gb, and the heterozygosity calculated as 0.39% and 0.5%, respectively. We generated 170.64 Gb (~107 × coverage) of Oxford Nanopore Technologies (ONT) long reads from *A. belladonna* and 828.66 Gb (~460 × coverage) of PacBio long reads from *D. stramonium* using the CCS model, which was subsequently corrected to 52.47 Gb (~29.15 × coverage) of high-fidelity (HiFi) reads (Table 1). We used these data to produce draft genome assemblies. The assemblies were polished using Illumina short reads (*A. belladonna*: 53.88 Gb; *D. stramonium*: 145.27 Gb) and then further improved to a chromosome-level assembly with Hi-C data (*A. belladonna*: 244.16 Gb; *D. stramonium*: 320.93 Gb) (Supplementary Tables 1 and 2). The final assembly of the *A. belladonna* genome was ~1.59 Gb, with a contig N50 size of 3.03 Mb and scaffold N50 size of 42.83 Mb, for which 99.50% of the sequences were anchored onto 36 pseudochromosomes (Supplementary Fig. 2a and Supplementary Table 3). The *D. stramonium* genome size was ~1.84 Gb, with a contig N50 size of 105.17 Mb, a scaffold N50 size of 156.09 Mb, 97.42% of the sequences anchored onto 12 pseudochromosomes (Supplementary Fig. 2b), and average pseudochromosome length of 149.77 Mb, among the pseudochromosomes, the shortest was 82.81 Mb (LG12) and the longest was 236.94 Mb (LG01) (Supplementary Fig. 2b and Supplementary Table 4). More than 99.63 and 99.94% of the genomic regions could be covered by short reads in the two species, indicating a high level of continuity of the assembly (Supplementary Fig. 3 and Supplementary Table 2).

Based on de novo and homology-based predictions and transcriptome data, 70,209 (*A. belladonna*) and 32,037 (*D. stramonium*) protein-coding genes were predicted with average lengths of 4802 and

**Table 1 | Summary of *A. belladonna* and *D. stramonium* genome assembly and annotation**

| Sequencing | A. belladonna | D. stramonium |
|---|---|---|
| Raw bases from WGS-Illumina (Gb) | 53.88 | 145.27 |
| Raw bases from WGS-Nanopore (Gb) | 170.64 | / |
| Raw bases from WGS-PacBio (Gb) | / | 828.66 |
| HiFi (Gb) | / | 52.47 |
| Raw bases from Hi-C (Gb) | 244.16 | 320.93 |
| **Assembly** | | |
| Number of contigs | 1753 | 1298 |
| Assembled genome size (bp) | 1,595,075,802 | 1,844,831,445 |
| Contig N50 (Mb) | 3.03 | 105.17 |
| Scaffold N50 (Mb) | 42.83 | 156.09 |
| Average Scaffold length (bp) | 1,468,823 | 1,503,559 |
| GC content (%) | 33.86 | 39.16 |
| Anchorage to chromosomes (%) | 99.50 | 97.42 |
| Complete BUSCOs (%) | 98.9 | 98.9 |
| **Annotation** | | |
| Percentage of repeat sequences (%) | 64.72 | 82.98 |
| LTR rate (%) | 43.14 | 66.16 |
| Number of predicted genes | 70,209 | 32,037 |
| Average gene length (bp) | 4801.59 | 3913.23 |
| Average CDS length (bp) | 1155.96 | 1127.27 |
| Average exon length (bp) | 288.83 | 235.26 |
| Average intron length (bp) | 756.68 | 734.77 |

3913 bp, and average coding sequence (CDS) lengths of 1156 and 1127 bp, respectively, and 98.9% of complete conserved homologs was observed in the two species based on BUSCO analysis (Table 1; Supplementary Tables 5–8). The spatial distribution of these protein-coding genes along the pseudochromosomes was uneven, with higher densities located at the ends of the chromosomal arms (Fig. 1). Comparison of gene structure with that of other species revealed that the average length and numbers of exons and introns were similar, while the gene length of *D. stramonium* was slightly shorter than that of *A. belladonna* (Supplementary Table 9). Moreover, 96.22 and 88.74% of all predicted protein-coding genes were annotated by at least one database (that is, SwissProt, TrEMBL, KOG, NR, GO, or KEGG) for *A. belladonna* and *D. stramonium*, respectively (Supplementary Table 10). In addition, 9333 and 27,434 noncoding RNA (ncRNA) genes were detected in the *A. belladonna* and *D. stramonium* genomes, yielding 437 and 173 microRNA (miRNA) genes, 2344 and 4880 transfer RNA (tRNA) genes, 3715 and 9326 small nuclear RNA (snRNA) genes and 2837 and 13,055 ribosomal RNA (rRNA) genes, respectively (Supplementary Table 11). Finally, the percentage of predicted repetitive elements was much higher in the genome of *D. stramonium*, 64.72% versus 82.98%, respectively (Table 1; Supplementary Tables 12–14).

We further sequenced 43.53 Gb of PacBio full-length cDNA data for *Lycium chinense*. After preprocessing by removing redundant reads from the generated data, 25,777 full-length transcripts were obtained (Supplementary Table 15). A total of 25,263 transcripts (genes) were identified in six databases: 24,761 in Nr, 24,925 in NT, 19,603 in GO, 13,567 in KEGG, 11,387 in KOG, 20,800 in SwissProt (Supplementary Table 16), and 72% of complete conserved homologs was observed by BUSCO analysis (Supplementary Table 17).

**Comparative genomic and phylogenomic analyses**

We clustered the annotated genes into gene families among *A. belladonna*, *D. stramonium*, and the other 10 species. A total of 318,352 genes from 12 species, including eight Solanaceae species, potato (*Solanum tuberosum*)[25], tomato (*Solanum lycopersicum*)[26], tobacco

(*Nicotiana tabacum*)[27], hot pepper (*Capsicum annuum*)[28], eggplant (*Solanum melongena*)[29], petunia (*Petunia axillaris*)[30], and four other angiosperms (Supplementary Table 18), were clustered into 37,515 gene families with an average of eight genes per family (Supplementary Table 19). A total of 43,921 *A. belladonna* genes were clustered into 23,879 gene families, and 26,109 *D. stramonium* genes were clustered into 20,328 gene families, which included 1172, and 657 specific families, respectively (Supplementary Fig. 4).

We selected 207 single-copy gene families among the 13 species (adding the full-length transcriptome data of *L. chinense*) to construct the phylogenetic tree, which showed that *A. belladonna* and *L. chinense* belong to a clade, while *D. stramonium*, *C. annuum*, *S. melongena*, *S. lycopersicum*, and *S. tuberosum* belong to the other clade. Based on molecular clock analysis, we estimated that *Petunia axillaris* diverged from the Solanaceae species ~43 million years ago (Mya) (33–56 Mya), *D. stramonium* diverged from the clade composed of *A. belladonna* and *L. chinense* ~28 Mya (23–37 Mya), and *A. belladonna* diverged from *L. chinense* ~24 Mya (18–31 Mya) (Fig. 2a).

We conducted expansion and contraction analysis based on the constructed phylogenetic tree and discovered 11,684 expanded and 1022 contracted families in *A. belladonna* compared with 1579 expanded and 5282 contracted families in *D. stramonium* (Fig. 2a). The GO enrichment analysis of the expanded gene families of *A. belladonna* suggested that these genes were enriched in "ATPase-coupled cation transmembrane transporter activity," "cellular nitrogen compound biosynthetic processes," and "regulation of flavonoid biosynthetic processes." The expanded gene families in *D. stramonium* were enriched in "oxidoreductase activity," "acting on NAD(P)H," "NADH dehydrogenase (ubiquinone) activity," "response to stimulus," and "defense response" (Supplementary Fig. 5; Supplementary Data 1, 2). KEGG functional enrichment analysis of the expanded gene families demonstrated that they were mainly assigned to "plant-pathogen interactions," "energy metabolism," "sesquiterpenoid and triterpenoid biosynthesis," and "flavonoid biosynthesis" (Supplementary Fig. 6; Supplementary Tables 20, 21).

**Two different forces drive genome size variation in *A. belladonna* and *D. stramonium***

Genome sizes vary widely across the Solanaceae, ranging from ~900 Mb in tomato to 4.5 Gb in tobacco and even larger in the Cyphomandra genus[31]. The proliferation of repetitive elements, especially the Gypsy family, was regarded as the main cause for the expansion of genomes of species in the Solanaceae family[32–36]. The genome sizes of *A. belladonna* and *D. stramonium* were larger than those of tomato and potato. To infer the driving force of the genome variation in *A. belladonna*, we used the distribution of synonymous substitution rates per gene ($K_S$) between collinear paralogous genes to identify whole-genome duplication events based on the assumption that the number of silent substitutions per site between two homologous sequences increases in a relatively linear manner with time. We selected a range of species for the comparative genomic investigation and assessment of polyploidization events in *A. belladonna* and *D. stramonium*: *S. lycopersicum*, *C. annuum*, *S. tuberosum*, and *S. melongena* as representatives of the Solanaceae, and *V. vinifera*, which represents the closest modern chromosome relative of the ancestral eudicot karyotype (AEK) with seven protochromosomes. By constructing the distribution of $K_S$ within each genome, we detected three and two polyploidization events in the genomes of *A. belladonna* and *D. stramonium*, respectively. The distribution of the reciprocal best hit (RBH) gene pair $K_S$ values exhibited a peak at ~0.55 in the Solanaceae species (*A. belladonna*, *D. stramonium*, *S. lycopersicum*, and *S. melongena*), further confirming the recent alpha WGT event common to all Solanaceae species. According to the relative times estimated by $K_S$ change, the Solanaceae species shared the older gamma paleopolyploidy event with other higher eudicots (Fig. 2b). Indeed, 716 syntenic

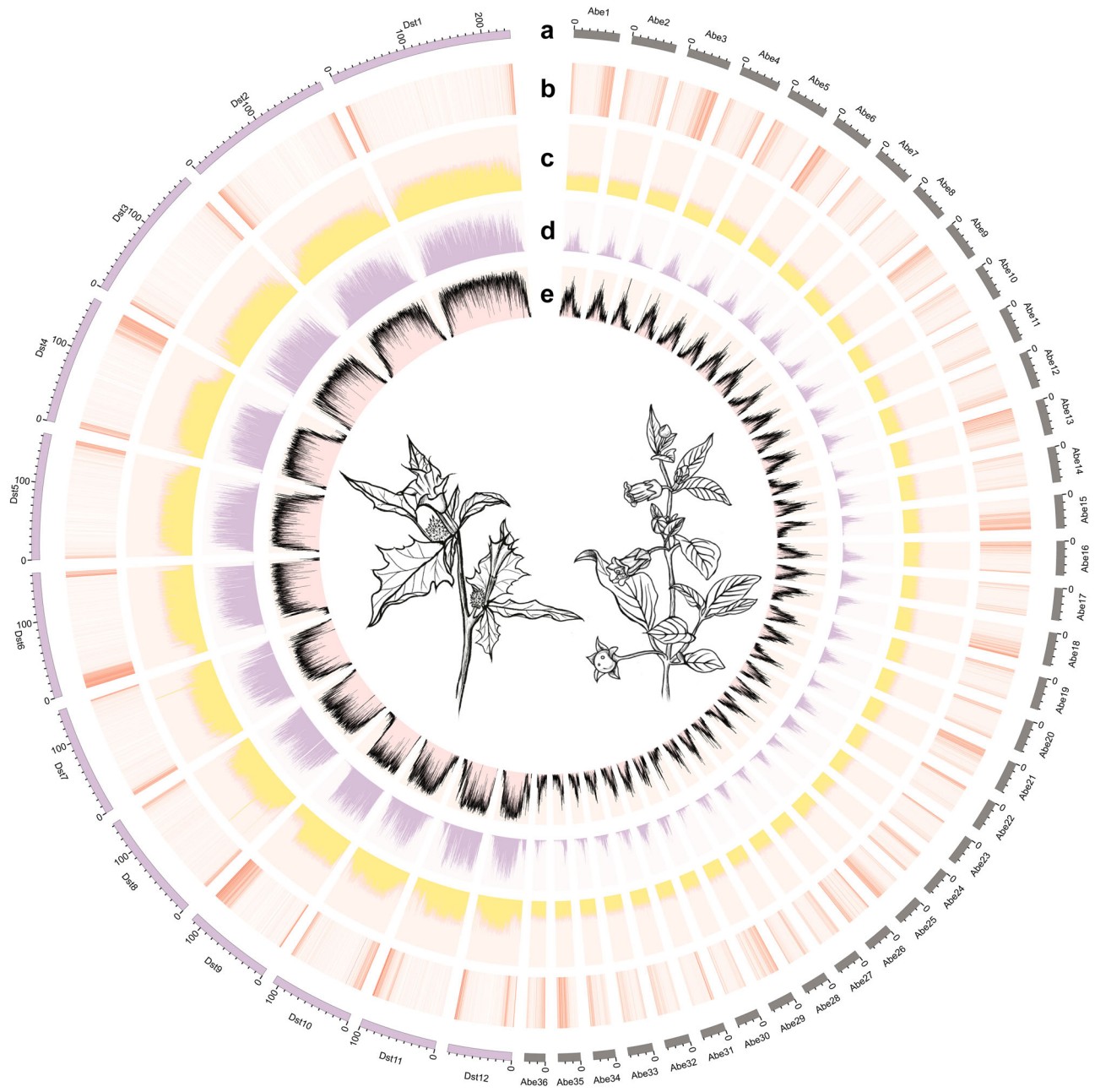

**Fig. 1 | An overview of the genomic features of *A. belladonna* and *D. stramonium*. a** The genomic landscape of the 36 *A. belladonna* pseudochromosomes (right) and 12 *D. stramonium* pseudochromosomes (left). All density information was counted in nonoverlapping 1-Mb windows. **b** Gene density. **c** Guanine-cytosine (GC) content. **d** The distribution of Copia-type retrotransposons. **e** The distribution of Gypsy-type retrotransposons.

blocks containing 28,463 paralogous gene pairs, were identified in the *A. belladonna* genome and the RBH paralog $K_S$ value distribution showed a peak at ~0.125–0.130, corresponding to another WGT event that occurred at ~17.50–18.20 MYA in *A. belladonna* after its split from other closely related species (Fig. 2b).

Synteny analyses between the genomes of *A. belladonna*, *D. stramonium*, and *V. vinifera* also showed clear evidence of another recent WGT event for *A. belladonna*. For each genomic region in *V. vinifera*, we typically found three matching regions in *D. stramonium* with a similar level of divergence and identified 3:1 syntenic depth ratios in the *A. belladonna-D. stramonium* genome comparison (Fig. 2c; Supplementary Figs. 7–9).

Given that the genome size of *D. stramonium* was ~250 Mb larger than the *A. belladonna* genome, we investigated the evolution of LTR retrotransposons and their potential contribution to the growth of the two species' genomes. We identified 832 Mb and 1,290 Mb (52.17% and 69.91%, respectively) of sequences in the assembled *A. belladonna* and *D. stramonium* genomes as transposable elements (TEs) (Supplementary Tables 13 and 14). The predominant type of TE was LTR elements, which represented ~667 and 1221 Mb (more than 64.65 and 79.73%, respectively) of the total repetitive sequence in the two genomes. Among the LTRs, most were Gypsy elements, which accounted for 38.92% and 72.91% of the total repetitive sequences in *A. belladonna* and *D. stramonium*, respectively, followed by Copia elements (22.57 and 5.17%, respectively) (Fig. 1d, e; Supplementary Fig. 10a; Supplementary Tables 13, 14). We then estimated the times of the LTR-RT burst in the two genomes, and the results suggested that the timing of the main LTR-RT burst was earlier in *A. belladonna* (~2.0 Mya, Supplementary Fig. 10b, c) while *D. stramonium* exhibited many more recent LTR-RT bursts (~0.6 Mya, Supplementary Fig. 10d, e).

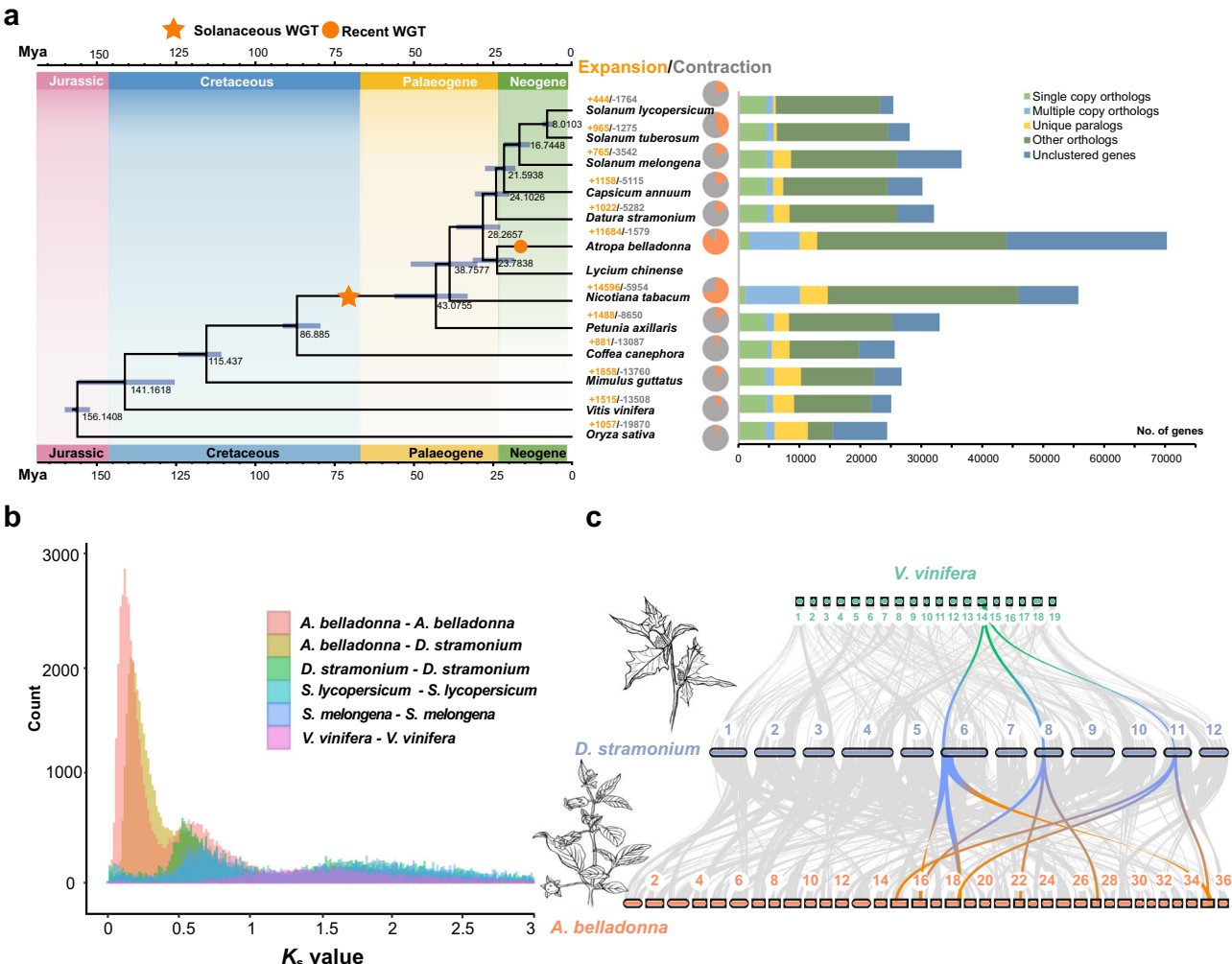

**Fig. 2 | Phylogenetic relationships and divergence times among 13 species.**
**a** Phylogenetic tree of *A. belladonna*, *D. stramonium*, and *L. chinense* along with 10 other plants. Gene families of the *A. belladonna*, *D. stramonium*, and other sequenced genomes are shown on the right. All branch bootstrap values are out of 100. Gene family expansions are indicated in orange, and gene family contractions are indicated in gray; the corresponding proportions of total changes are shown using the same colors in the pie charts. The estimated divergence time (million years ago, Mya) is indicated at each node; bars are the 95% highest posterior density (HPD). Circles in orange represent recent whole-genome duplication events. **b** *K*s values revealed a recent WGT event during the evolution of *A. belladonna*, a WGT event shared by Solanaceae species, and a WGT event shared by Solanaceae species and *V. vinifera*. **c** Collinearity between *A. belladonna*, *D. stramonium*, and *V. vinifera* chromosomes. The collinearity pattern shows that a typical ancestral region in the *V. vinifera* genome can be traced to three regions in the *D. stramonium* genome and nine regions in the *A. belladonna* genome. Gray wedges in the background indicate syntenic blocks spanning more than 15 genes between the genomes.

## *D. stramonium* and *A. belladonna* employ a conserved biosynthetic pathway to produce mTAs

Given that both *D. stramonium* and *A. belladonna* produce mTAs, it is of interest to investigate whether *D. stramonium* employs the same biosynthetic pathway as *A. belladonna* to produce mTAs. To this end, we first elucidated the TAs biosynthetic pathway by transgenic and biochemical approaches in *A. belladonna* to identify *MPO* and *TRI*, which are the only two uncharacterized TAs biosynthetic genes in this species. The oxidative deamination of *N*-methylputrescine is a key step in TAs biosynthesis, and this reaction may be catalyzed by *N*-methylputrescine oxidase (MPO). However, the gene encoding MPO has not been characterized in TAs-producing plants. We retrieved three putative MPOs, EVM0017027.2, EVM0068072.1, and EVM0064643.2, using BLASTP search in which MPO from tobacco was used as a query. However, we cloned only two of them from *A. belladonna*, *EVM0017027.2* and *EVM0068072.1*, which were named *AbMPO1* and *AbMPO2*, respectively. *AbMPO1* was highly expressed in the roots of *A. belladonna*, while *AbMPO2* was predominantly expressed in aboveground tissues (Supplementary Fig. 11a). Suppressing the expression of *AbMPO1* but not *AbMPO2* significantly decreased the contents of

hyoscyamine and scopolamine in hairy root cultures of *A. belladonna* (Fig. 3b and Supplementary Fig. 11). Thus, AbMPO1 is the primary functional MPO involved in mTAs biosynthesis in *A. belladonna*. We also identified the TRI in *A. belladonna* via in vitro TRI enzyme activity assays (see below). Subsequently, we used the functionally characterized genes of *A. belladonna* as seeds to retrieve all homologous genes across nine Solanaceae for each gene family involved in TAs/mTAs biosynthesis. By constructing the phylogenetic trees of each family at each step (Supplementary Figs. 12–23), we defined the well-supported subfamily containing the functionally tested sequence as the most likely mTAs-related group at each step. Subsequently, we determined the mTAs biosynthetic genes of each species and constructed phylogenetic relationships of all homologs across all species for each subfamily (Supplementary Figs. 24, 25). Most of the genes in the upstream pathway of mTAs biosynthesis were multicopy and conserved across Solanaceae, for which the phylogenetic relationships were consistent with the interspecific relationships. Moreover, the results of microsynteny analysis were consistent with the results of phylogenetic analysis. All genes in modules I and II showed a very high degree of synteny across Solanaceae with high sequence similarity except PPAR

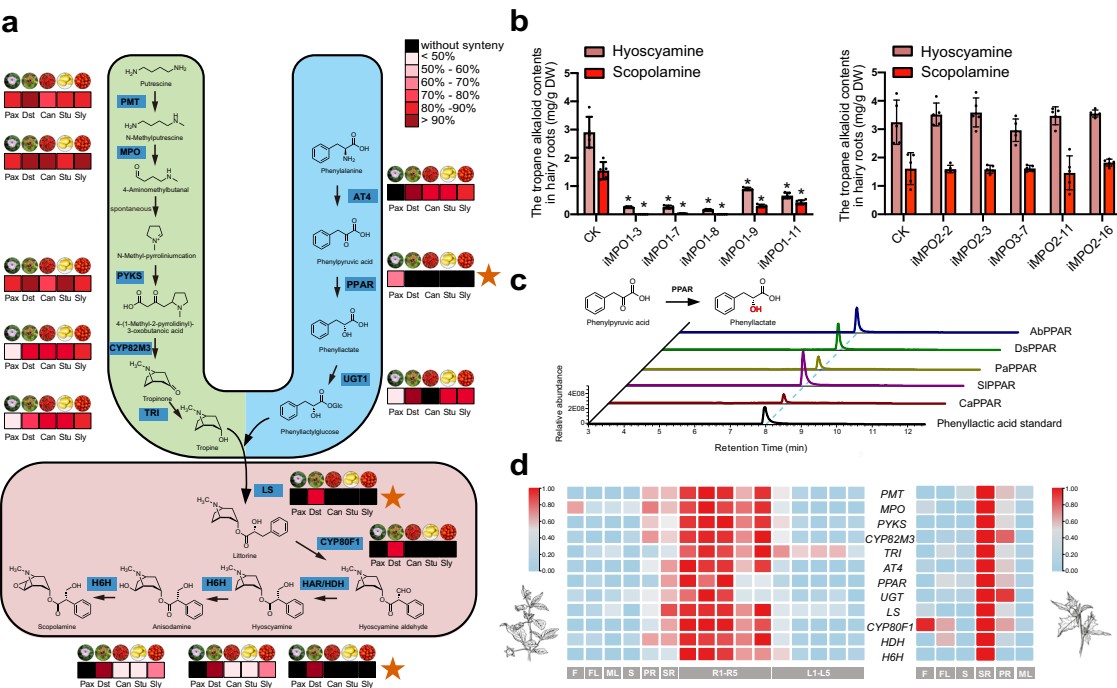

**Fig. 3 | Biosynthetic pathway of medicinal tropane alkaloids (mTAs).**
**a** Schematic representation of the medicinal tropane alkaloid biosynthetic pathway. Light green represents module I, including PMT putrescine *N*-methyltransferase, MPO *N*-methylputrescine oxidase, PYKS type III polyketide synthase, CYP82M3 tropinone synthase, and TRI tropinone reductase I. Light blue represents module II, including AT4 aromatic amino acid aminotransferase 4, PPAR phenylpyruvic acid reductase, and UGT1 phenyllactate UDP-glycosyltransferase. Light red represents module III, including LS littorine synthase, CYP80F1 littorine mutase, HDH hyoscyamine dehydrogenase, and H6H hyoscyamine 6β-hydroxylase. Pax, *P. axillaris*; Dst, *D. stramonium*; Can, *C. annuum*; Stu, *S. tuberosum*; Sly, *S. lycopersicum*. **b** The functional characterization of MPOs from *A. belladonna* by RNA interference in hairy roots. Left, the hyoscyamine and scopolamine contents in AbMPO1-RNAi root cultures. Right, the hyoscyamine and scopolamine contents in AbMPO2-RNAi root cultures. CK, control root cultures. iMPO1, hairy root cultures with *AbMPO1* RNAi. iMPO2, hairy root cultures with *AbMPO2* RNAi. DW, dry weight. The data are presented as means values +/− s.d. (*n* = 5 biologically independent samples). *, represents a significant difference from control line (CK) at the levels of

*P* < 0.01 as determined by two-sided Student's *t*-test. For hyoscyamine contents, **P* = 0.0000 (iMPO1-3), **P* = 0.0000 (iMPO1-7), **P* = 0.0000 (iMPO1-8), **P* = 0.0000 (iMPO1-9), and **P* = 0.0000 (iMPO1-11). For scopolamine contents, **P* = 0.0000 (iMPO1-3), **P* = 0.0000 (iMPO1-7), **P* = 0.0000 (iMPO1-8), **P* = 0.0000 (iMPO1-9), and **P* = 0.0000 (iMPO1-11). **c** Extracted ion chromatograms showing the in vitro catalytic activity of five purified recombinant PPARs from *P. axillaris* (PaPPAR, Peaxi162Scf00071g00082.1), *A. belladonna* (AbPPAR, EVM0020147.2), *D. stramonium* (DsPPAR, DstT013938.1), *C. annuum* (CaPPAR, XM_016708230.2), and *S. lycopersicum* (SlPPAR, XM_004229823.4) with phenylpyruvic acid (PPA) used as a substrate. The blue dotted line represents the retention time of phenyllactic acid, the product of PPAR. **d** Gene expression profiles (in normalized TPMs) of different tissues in two species are presented in the heatmap alongside the gene names (F fruit, FL flower, L mature leaf, S stem, PR primary root, SR secondary roots, R1-R5 the roots of *A. belladonna* at five different development stages, L1-L5 the young leaves of *A. belladonna* at five different development stages). Source data underlying **b** and **d** are provided as a Source Data file.

(Fig. 3a; Supplementary Figs. 26–38). Recently, a widespread alternative trans-cinnamic acid (CA) formation pathway in plants was reported. Two PPARs from tea plants were also found to transform phenylpyruvic acid into phenyllactic acid[37]. Thus, we hypothesized that this enzyme is also widely distributed across Solanaceae and is unlikely to be a bottleneck in the biosynthesis of phenyllactylglucose. To test this hypothesis, we characterized the function of PPARs from *A. belladonna*, *D. stramonium*, *P. axillaris*, *S. lycopersicum*, and *C. annuum* by in vitro enzymatic assays. Consistent with our speculation, all the tested PPARs transformed phenylpyruvic acid into phenyllactic acid (Fig. 3c; Supplementary Fig. 39; Supplementary Table 22). Taken together, the identified scopolamine biosynthetic genes and results of phylogenetic analysis combined with microsynteny analysis revealed the conservation of phenyllactylglucose and tropine biosynthetic genes in Solanaceae species.

Interestingly, the phylogenetic analysis showed that the *TRI, LS, CYP80F1, HDH,* and *H6H* genes of *A. belladonna* and *D. stramonium* are clustered on one branch and divergent from related genes in other Solanaceae species without mTAs (Supplementary Figs. 24, 25). The sequence similarity of the *TRI, LS, CYP80F1, HDH,* and *H6H* genes between *A. belladonna* and *D. stramonium* were likewise dramatically higher than those between *A. belladonna* and other species (Supplementary Fig. 40). Furthermore, the microsynteny analysis showed that

the *LS, CYP80F1,* and *HDH* genes in *A. belladonna* have syntenic genes only in *D. stramonium* (Fig. 3a; Supplementary Figs. 34–37). The integrated evidence of multiple resources from synteny analysis, phylogeny analysis and sequence alignments indicated that *D. stramonium* employs the same biosynthetic route as *A. belladonna* to produce mTAs, although the two species belong to distant genera.

Using RNA-Seq reads generated from tissues in two species with mTAs, we further examined the expression of all the genes identified above. The results showed that at least one mTAs-related gene identified at each biosynthesis step was highly expressed in the roots of *A. belladonna* and *D. stramonium* (Fig. 3d). The high levels of hyoscyamine and scopolamine in the roots of *A. belladonna* and *D. stramonium* are likely attributable to the constant and high expression of these genes. To examine the coexpression of the pathway-related genes more broadly, we constructed weighted gene coexpression networks by WGCNA using all differentially expressed genes in *A. belladonna*, obtaining seven clusters (Supplementary Fig. 41, Supplementary Data 3–9). Remarkably, the identified probable mTAs genes are classified into 'blue module' and are highly expressed in root tissues with other related genes (Supplementary Figs. 42, 43b). The GO enrichment analysis of these 'blue module genes' suggested that they are enriched in the "tropane alkaloid biosynthetic process," "tropane alkaloid metabolic process," "alkaloid metabolic process," "oxidoreductase

activity, acting on peroxide as acceptor," and "peroxidase activity" (Supplementary Fig. 43a).

## The evolution of gene clusters correlates with the widespread distribution of TAs across Solanaceae

Given the wide distribution of TAs across Solanaceae and the high conservation of tropine biosynthetic genes, we further examined the chromosomal positions of those genes (Supplementary Figs. 44, 45; Supplementary Tables 23, 24) and found that *CYP82M3* and *TRI* genes were clustered in representative genomes of Solanaceae species except tobacco (Fig. 4a). CYP82M3 catalyzes the formation of tropinone; TRI subsequently reduces tropinone to produce tropine, the core structure of TAs, which has been documented in many Solanaceae species[3,4]. We speculated that the evolution of this gene cluster contributed to the widespread distribution of tropine in this family. Moreover, this gene cluster cannot be identified in the most distantly related species in our analysis, *Ipomoea triloba* (Convolvulaceae). We proposed that the recruitment of *CYP82M3* and *TRI* occurred after the ancient WGT event in Solanaceae because no intraspecies synteny region was found in any analyzed species except *A. belladonna*, in which a recent WGT event occurred (Fig. 4a).

We noticed that only one *CYP82M3* gene existed in the gene cluster across Solanaceae; nevertheless, *TRI* underwent tandem duplication (in *C. annuum* and *D. stramonium*) and/or duplication caused by WGT in *A. belladonna* (Fig. 4a). This led us to hypothesize that the duplication of *TRI* contributed to the distribution of tropine biosynthesis in Solanaceae species. To test this hypothesis, we first tested for tropine in the species and detected it in *C. annuum*, *A. belladonna* and *D. stramonium* but not in *P. axillaris*, *S. lycopersicum*, and *S. tuberosum* (Fig. 4b). Moreover, the duplicated *TRI* genes showed a high diversity of tissue expression patterns, while all the *CYP82M3* genes in the species investigated in this study were predominantly expressed in the root (Fig. 3d; Supplementary Fig. 46). Contrary to the result in non-TAs-producing species, at least one TRI gene was highly expressed in the roots of TAs-producing species, suggesting that the subfunctionalization of *TRI* occurred after its duplication. Biochemical assays further confirmed that the plants with *TRI* duplications harbored at least one functional *TRI* with tropinone reductase activity (Fig. 4c). However, in contrast with the result that we could not detect tropine in tomato, TRI from tomato also showed TRI activity. This prompted us to conduct further enzymatic kinetic analysis of TRI from plants with *TRI* duplication (*A. belladonna*, *D. stramonium*, and *C. annuum*) and without *TRI* duplication (*S. lycopersicum*). The $K_{cat}/K_m$ value of SlTRI was dramatically low, only -0.01, 0.006, and 0.11 of that of AbTRI, DsTRI, and CaTRI, respectively (Fig. 4d; Supplementary Fig. 47). Moreover, the $K_m$ value of SlTRI was -43.56, 7.88, and 36.09 times greater than that of AbTRI, DsTRI, and CaTRI, respectively (Fig. 4e; Supplementary Fig. 47). The enzymatic kinetic analysis indicated that the catalytic efficiency of SlTRI is dramatically lower than that of AbTRI, DsTRI, and CaTRI. Since the crystal structures of DsTRI have been resolved[38], we compared the amino acid sequences of TRIs from these five species, especially the ten residues that were predicted to contact with tropinone. Only one of the ten residues could be correlated with the low or null catalytic efficiency of SlTRI and StTRI (Supplementary Fig. 48). Val[168] was conserved across the TAs-producing species but substituted by Leu[159] in SlTRI and StTRI. Consequently, we analyzed the enzymatic kinetics of SlTRI[L159V] and StTRI[L159V]. The $K_{cat}/K_m$ value of SlTRI[L159V] increased, followed by a decrease in the $K_m$ value (Fig. 4c–f; Supplementary Fig. 47). More significantly, the StTRI[L159V] gained the ability to catalyze tropinone to tropine (Fig. 4c–f, Supplementary Fig. 47). In addition, we tested the function of CYP82M3 from *C. annuum* and *S. lycopersicum* in tobacco leaves. Both CaCYP82M3 and SlCYP82M3 transformed 4-(1-methyl-2-pyrrodinyl)-3-oxobutanoic acid into tropinone when they were coexpressed with AbPMT and AbPYKS in tobacco leaves, indicating that the

enzyme activity of CYP82M3 was conserved in Solanaceae and that the failure to detect tropine in tomato was not because of the functional deficiency of CYP82M3 (Fig. 4d). Together, these results suggested that the genes involved in the two important tropine biosynthetic steps were arranged in a gene cluster that is conserved across Solanaceae. The subfunctionalization of *TRI* after its duplication promoted higher expression levels in roots and more robust TRI enzymatic activity.

## Evolution of mTAs biosynthetic pathway-specific genes

Although TAs are widely distributed across Solanaceae, mTAs, such as hyoscyamine and scopolamine, can be detected only in limited tribes, especially in Daturaeae and Hyoscyameae. Compared with upstream pathway genes that were conserved in Solanaceae, mTAs biosynthetic pathway-specific genes, such as *LS*, *CYP80F1*, and *HAR*, from the two mTAs producing species, *A. belladonna* and *D. stramonium*, formed a clade clearly distinct from other homologs of the species without mTAs (Supplementary Fig. 25). Consistently, we observed littorine, the direct product of LS, only in *A. belladonna* and *D. stramonium* (Fig. 5a). We speculated that the functional LS only existed in Daturaeae and Hyoscyameae. To test this hypothesis, we carried out relative enzyme activity assays of three LS homologs in *D. stramonium* (DsLS), *A. belladonna* (AbLS), and *L. chinense* (LcLS) by expressing them transiently in tobacco leaves together with AbUGT1, which converts phenyllactate into phenyllactylglucose used as substrate. After infiltration with phenyllactate and tropine, we detected littorine in tobacco leaves expressing DsLS and AbLS but not in those expressing LcLS. This suggested that both DsLS and AbLS have LS activity (Fig. 5b, c). In addition, DsLS had significantly higher enzyme activity than AbLS in our test, suggesting that DsLS is a better candidate for engineering mTAs in plants or in yeast (Fig. 5c). To understand *LS* gene evolution, we employed microsynteny analysis, which revealed that the syntenic region of *LS* in *A. belladonna* showed a high degree of synteny among all tested species, but *LS* genes were dramatically absent in *P. axillaris*, *C. annuum*, *S. lycopersicum*, and *S. tuberosum* (Supplementary Fig. 34). Interestingly, by performing more detailed sequence analysis, we found that the portion of the *AbLS* promoter region was highly conserved in all the tested non-mTAs-producing species, and even the first two exons could be detected in *Iochroma cyaneum*, a species in Physalideae, with high similarity (Supplementary Figs. 34, 35). Combining the results of microsynteny analysis with those of phylogeny analysis, sequence alignments and enzyme activity assays led to the hypothesis that mTAs biosynthetic genes have been lost in non-mTAs-producing species. To address this, we further investigated the microsynteny of *CYP80F1*. Consistent with the observation for *LS*, the genomic region in which *CYP80F1* is located also showed a high degree of synteny among the tested species and *CYP80F1* was absent in non-mTAs-producing species again (Fig. 5d). Nevertheless, a syntenic *CYP80F1* homolog was detected in petunia. Moreover, two traces of the first exon of *CYP80F1* and a trace of the second exon of *CYP80F1* were detected in the syntenic region of *C. annuum* and *S. tuberosum*, respectively (Fig. 5d). Thus, our evidence suggested that the gene loss in non-mTAs-producing tribes contributed to the limited distribution of mTAs in Solanaceae.

## A cytochrome P450 in the CYP82M subfamily accounts for the N-demethylation of hyoscyamine

Although the biosynthesis of hyoscyamine and scopolamine has been resolved, the enzymes modifying them have not been reported. Norhyoscyamine, the product of N-demethylation of hyoscyamine, was identified in many mTAs-producing species, such as *A. myoporoides*, *A. pannosa* and *A. walcottii* in Anthotroche, *Datura arborea* in *Datura* and the *Anthrocercis albicans × Duboisia myoporoides* hybrid[3]. To characterize the enzymes involved in the N-demethylation of hyoscyamine, we first measured the contents of norhyoscyamine in different tissues

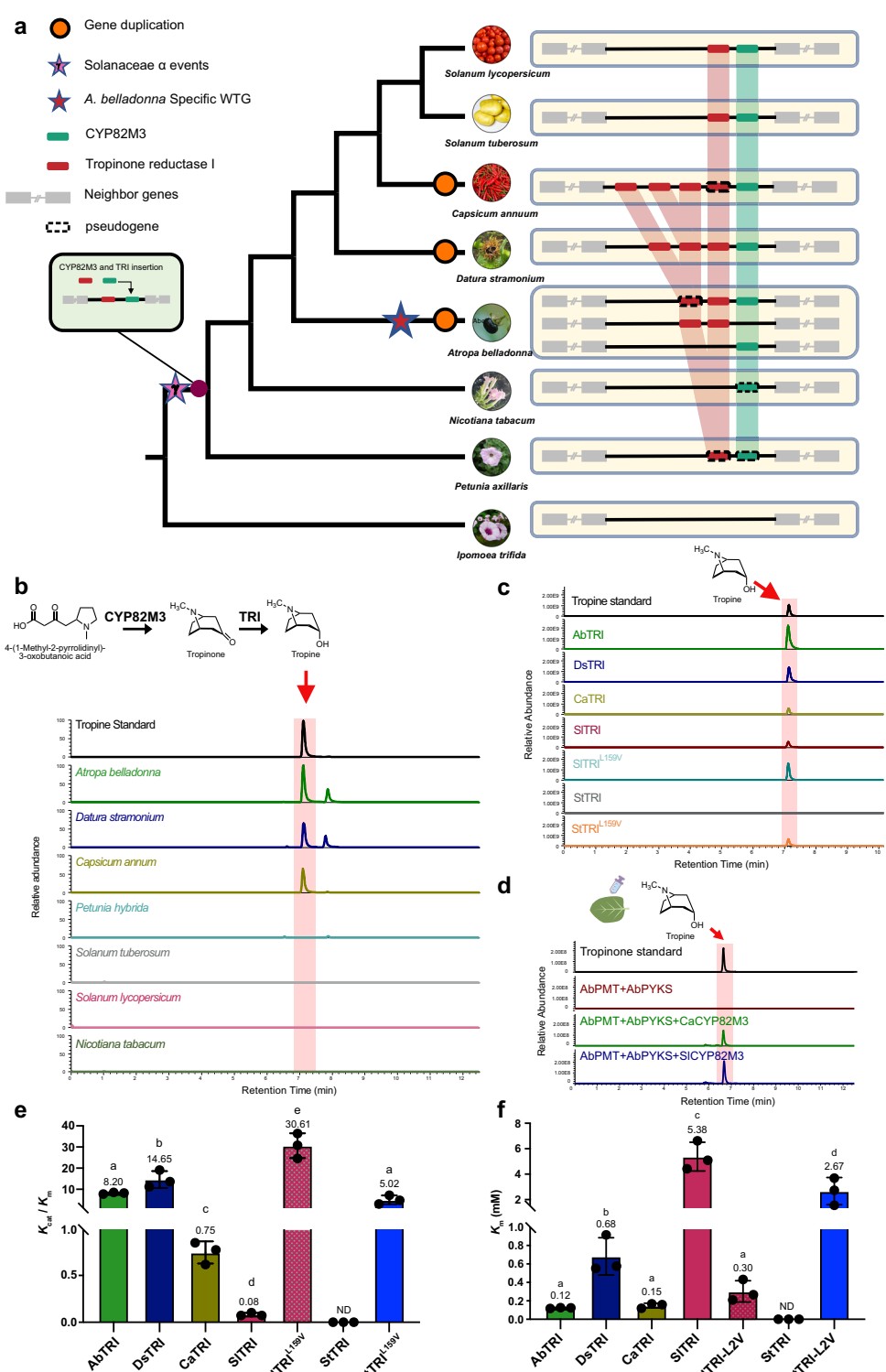

**Fig. 4 | The evolution of the tropine gene cluster is associated with tropine diversity across the Solanaceae family. a** The tropine gene cluster syntenic regions of seven Solanaceae species and the outgroup species *Ipomea trifida* (Convolvulaceae). Genes encoding TRI are indicated in the red rectangle, and genes encoding CYP82M3 are indicated in the green rectangle. **b** Analysis of tropine in plants by combined LC/MS. The roots, stems and leaves of plants were analyzed for the detection of tropine. The extracted ion chromatograms of tropine are highlighted in red. **c** Extracted ion chromatograms showing the in vitro activity of seven purified recombinant TRIs from *A. belladonna* (AbTRI), *D. stramonium* (DsTRI), *C. annuum* (CaTRI), *S. lycopersicum* (SlTRI), *S. tubersoum* (StTRI), SlTRI[L169V], and

StTRI[L169V] with tropinone used as substrate. **d** Extracted ion chromatograms showing the in vivo activity of CaCYP82M3 and SlCYP82M3 in tobacco leaves. **e, f** The evaluation of enzymatic kinetics on five wild-type TRIs and two mutants by $K_{cat} / K_m$ values (**e**) or by $K_m$ values (**f**). The raw Michaelis–Menten curves of these TRIs for the NADPH-dependent reduction reaction of tropinone are presented in Fig. S31. The data are presented as means values +/− s.d. ($n = 3$ biologically independent samples). Different letters in **e** and **f** indicate significantly different values compared with each other at $P < 0.05$ analyzed by two-sided Student's *t*-test. ND not detectable. Source data underlying **e** and **f** are provided as a Source Data file.

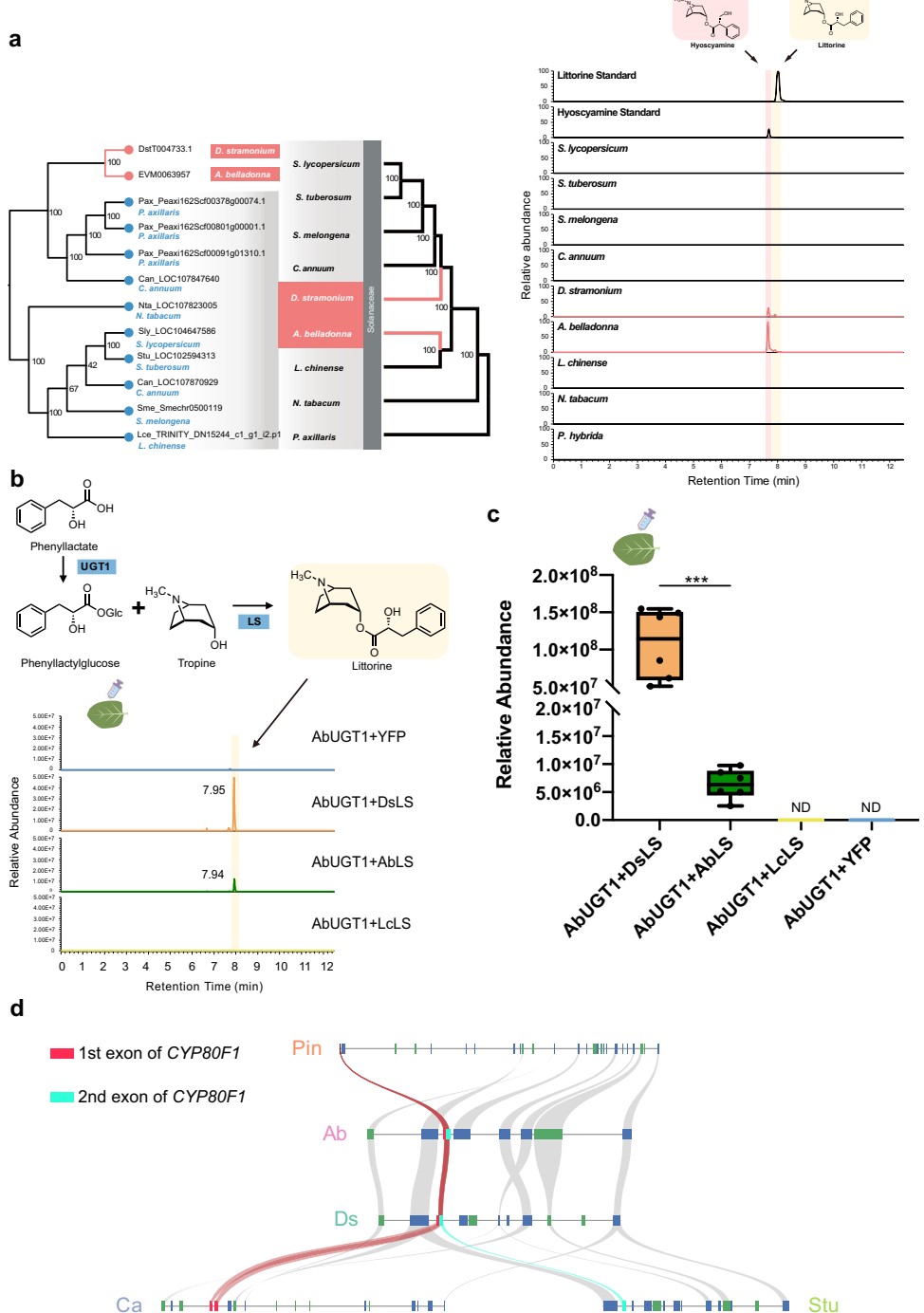

**Fig. 5 | The evolution of mTAs-specific genes. a** Left, the gene tree of *LS* genes and species tree of nine Solanaceae species; right, the extracted ion chromatograms of hyoscyamine (light red) and littorine (light yellow). The roots, stems, and leaves of plants were analyzed for the detection of hyoscyamine and littorine by UPLC-MS. **b** Extracted ion chromatograms showing the in vivo activity of three LS form *A. belladonna* (AbLS), *D. stramonium* (DsLS) and *L. chinense* (LcLS) by coexpressing AbUGT, which converted phenyllactate into phenyllactylglucose in tobacco leaves; phenyllactate and tropine were used as substrates by coinfiltration into tobacco leaves. **c** The relative LS activity of AbLS and DsLS. For the boxplot (*n* = 6 biologically independent samples), the centerline, median; box limits, upper and lower quartiles; whiskers, data range. *** represented significant difference analyzed by two-sided Student's *t*-test at the level of *P* < 0.001. ***P = 0.0004 (AbUGT1+DsLS compared with AbUGT1 + AbLS). ND not detectable. **d** Synteny analysis of the *CYP80F1* gene in different tribes. Pin, *P. inflata*; Ab, *A. belladonna*; Ds, *D. stramonium*; Ca, *C. annuum*; Stu, *S. tuberosum*. Syntenic pieces of *CYP80F1* are highlighted in red. The red rectangle on the chromosome represents the first exon of *CYP80F1*; the light blue rectangle on the chromosome represents the second exon of *CYP80F1*. Source data underlying **c** is provided as a Source Data file.

of *A. belladonna*. Consistent with the distribution of hyoscyamine, norhyoscyamine was predominantly distributed in the roots of *A. belladonna*, especially in the primary roots (Fig. 6a, b). Because members of the CYP82 subfamily from tobacco convert nicotine to nornicotine by *N*-demethylation, we first retrieved all *CYP* genes using

hidden Markov model-based conserved motif searches in *A. belladonna* and *D. stramonium* (Fig. 6c). By building a phylogenetic tree of CYPs in *A. belladonna*, *D. stramonium* and *Arabidopsis*, we found that eight CYPs from *A. belladonna*, including two AbCYP82M3s, were clustered in the CYP82M clade. Further coexpression analysis revealed

that one member, *EVM0022661.2*, was highly coexpressed with *AbCYP82M3*s in the primary root and secondary root of *A. belladonna*. To test the enzymatic function of EVM0022661.2, we cloned *EVM0022661.2* and expressed it transiently in *N. benthamiana* leaves. After infiltration with hyoscyamine as the substrate, norhyoscyamine was detected in tobacco leaves expressing EVM0022661.2 (Fig. 6e), suggesting that EVM0022661.2 has hyoscyamine *N*-demethylase activity. To further verify the activity of EVM0022661.2, we expressed EVM0022661.2 in yeast and the corresponding *N*-demethylase activity was assayed in yeast microsomes (Fig. 6f). Norhyoscyamine was readily detected in assays containing microsomes isolated from yeast expressing EVM0022661.2 and hyoscyamine. To validate the function of EVM0022661.2 in plants, we overexpressed it in the hairy roots of *A. belladonna* (Supplementary Fig. 49). The norhyoscyamine levels in *EVM0022661.2* overexpression lines significantly increased by ~2.09-fold to ~5.70-fold, compared with those in vector control lines (Fig. 6g). Furthermore, we applied the CRISPR/Cas9 system to generate mutants of the *EVM0022661.2* gene in the hairy roots of *A. belladonna* (Supplementary Fig. 49). When *EVM0022661.2* was edited, the norhyoscyamine level decreased to ~17% to ~26% of that in vector control lines (Fig. 6h). Taken together, our data provide evidence that the production of norhyoscyamine is catalyzed by EVM0022661.2.

## Discussion

Some species in the family Solanaceae are of great value to human, such as tomato and potato, while others produce such lethal toxic compounds that they are given ominous names, such as *A. belladonna*, which is also called deadly nightshade. Given that most sequenced genomes in the Solanaceae are from crops, the high-quality genomes of the two medical plants presented here have provided a great opportunity to gain a complete understanding of the evolution of the mTAs biosynthetic pathway. Based on the high-quality genome sequence, we identified an additional WGT event for *A. belladonna* after its split from *L. chinensis*. The genome size of *A. belladonna* is smaller than that of *D. stramonium*; however, the gene number in *A. belladonna* is much larger than that in *D. stramonium* due to polyploidy. We found that extreme TE amplification (especially of Gypsy TEs) in *D. stramonium* caused this increase in genome size.

Although the chemical and enzymatic routes of hyoscyamine and scopolamine biosynthesis have been clarified, the genes encoding those enzymes have not all been characterized in a single species. In the present work, we elucidated the hyoscyamine and scopolamine biosynthetic pathways by functional identification of *MPO* and *TRI* from *A. belladonna*. Thus, the mTAs biosynthetic genes of *A. belladonna* have been functionally characterized, which will facilitate the exploration of mTAs evolution. Furthermore, combined with the high-quality genome, abundant transcriptome data, easy genetic manipulation, and easy virus-induced gene silencing, it also makes *A. belladonna* an ideal reference species for studying the further modification of mTAs, transcriptional regulation, and biosynthesis of other mTAs. Correspondingly, we characterized a *N*-demethylase in *A. belladonna* that converts hyoscyamine to norhyoscyamine (Fig. 6). However, trace amounts of norhyoscyamine were detected when we performed an *N*-demethylation activity test in transient expression assay in tobacco leaves and in yeast microsomes (Fig. 6d, e). This may be due to non-specific enzyme catalysis because *N*-demethylation of TAs might be a general mechanism to detoxify metabolites, even in mammals[39,40]. Consistently, norhyoscyamine was also detected in the engineered hyoscyamine-producing yeast strains[41]. Furthermore, two CYPs from humans, HsCYP2C19 and HsCYP2D6, involved in the *N*-demethylation of hyoscyamine were identified[41]. Considering the distant relationship between humans and *A. belladonna* and the low identity between HsCYP2C19/HsCYP2D6 and EVM0022661.2 (lower than 25%, Supplementary Fig. 50), we speculate that the similar function of *N*-demethylation of hyoscyamine is the consequence of convergent evolution.

Solanaceae species produce diverse types of TAs in addition to hyoscyamine and scopolamine. However, the genetic basis of the broad distribution of TAs across Solanaceae remains unclear. Using the abundant genomic data for the Solanaceae family, we explored the evolution of the biosynthesis of tropine, the core structural component of TAs. Through phylogenetic analysis, microsynteny analysis, and in vivo/in vitro enzymatic assays, we provided a clear evolutionary landscape of TAs biosynthesis in the Solanaceae family. Our results suggested that the biosynthetic genes of tropine are conserved across this family. Interestingly, tropine was detected in hot pepper, a popular vegetable and source of capsaicin. Although the poisonousness of tropine has not been reported, we cannot exclude the possibility that tropine in hot pepper can be converted into physiologically toxic compounds. Furthermore, we identified a conserved gene cluster comprising *CYP82M3* and *TRI* in Solanaceae. Recently, much evidence has indicated that the gene cluster is more common than previously thought, especially in the biosynthesis of plant natural products, such as morphinan[42], thalianol[43] and momilactones[44]. In Solanaceae, Itkin et al. reported a conserved gene cluster contributing to steroidal glycoalkaloid (SGA) biosynthesis in potato and tomato[45]; Fan et al. reported a gene cluster associated with medium chain acyl sugar biosynthesis, which was located next to the SGAs gene cluster[46], implying that this mechanism might be prevalent in this family and helping to elucidate the biosynthesis and evolution of other secondary metabolisms in Solanaceae. By linking the TRI-CYP82M3 gene cluster genotype and tropine phenotype, we found that this gene cluster cannot ensure tropine production (Fig. 4), which is contracted with the synchronization of the gene cluster genotype and metabolite phenotype observed in other species. Both tomato and potato contain this gene cluster, but tropine was not detected in them (Fig. 4b). In contrast, tropine production is associated with the duplication of *TRI*, suggesting that more powerful TRI with higher tropinone reductase activity and more specific expression in roots emerged through TRI duplication in some genera (Fig. 3). However, we cannot exclude the possibility that the duplication of TRI occurred in Solanoideae and has since been lost in some genera in Solaneae. In *Physalis floridana*[34], a species belonging to Physaleae, also has a duplicated TRI in its TRI-CYP82M3 gene cluster (Supplementary Fig. 51). Furthermore, it is very likely that *Physalis floridana* contains TAs as well because *Physalis peruviana*, another species in the *Physalis* genus, contains tropine[47]. It would be interesting to investigate TAs content and biosynthesis in *P. floridana* in the future. Regarding the retained *TRI* in tomato and potato, one possibility is that they are undergoing nonfunctionalization or pseudogenization, similar to what happened to *LS* and *CYP80F1* in *C. annuum*. Nevertheless, we cannot reject the speculation that *TRI*s in tomato and potato have undergone neofunctionalization and may have distinct biochemical activities by gaining novel enzymatic functions that have not been identified. Taken together, our findings indicated that gene clusters combined with gene duplication underlie the widespread distribution of tropine in Solanaceae.

Regarding the specific esterification and consequent modification of tropine to produce scopolamine in two distant genera, our results support previous findings that these two species (representatives of Daturaeae and Hyoscyameae) do not comprise a monophyletic clade (Fig. 2a). It is likely that the biosynthesis of mTAs appeared more than once, even in the same family, because of convergent evolution. The microsynteny analysis rejected this hypothesis, although the results of phylogenetic analysis and sequence alignments supported it. First, the genomic region *LS* showed a high degree of synteny across Solanaceae, even in *Petunia*, which is not included in the "x = 12" clade; the nucleotide sequence identity between *AbLS* and *DsLS* is as high as ~89%, and the functional *LS* of the two species are syntenic (Supplementary Figs. 34, 40). Second, a portion of the *AbLS* promoter region can be traced to the corresponding syntenic region of species in *Petunia*, *Capsicum*, *Solanum*, and *Iochroma*. Even the first two exons of

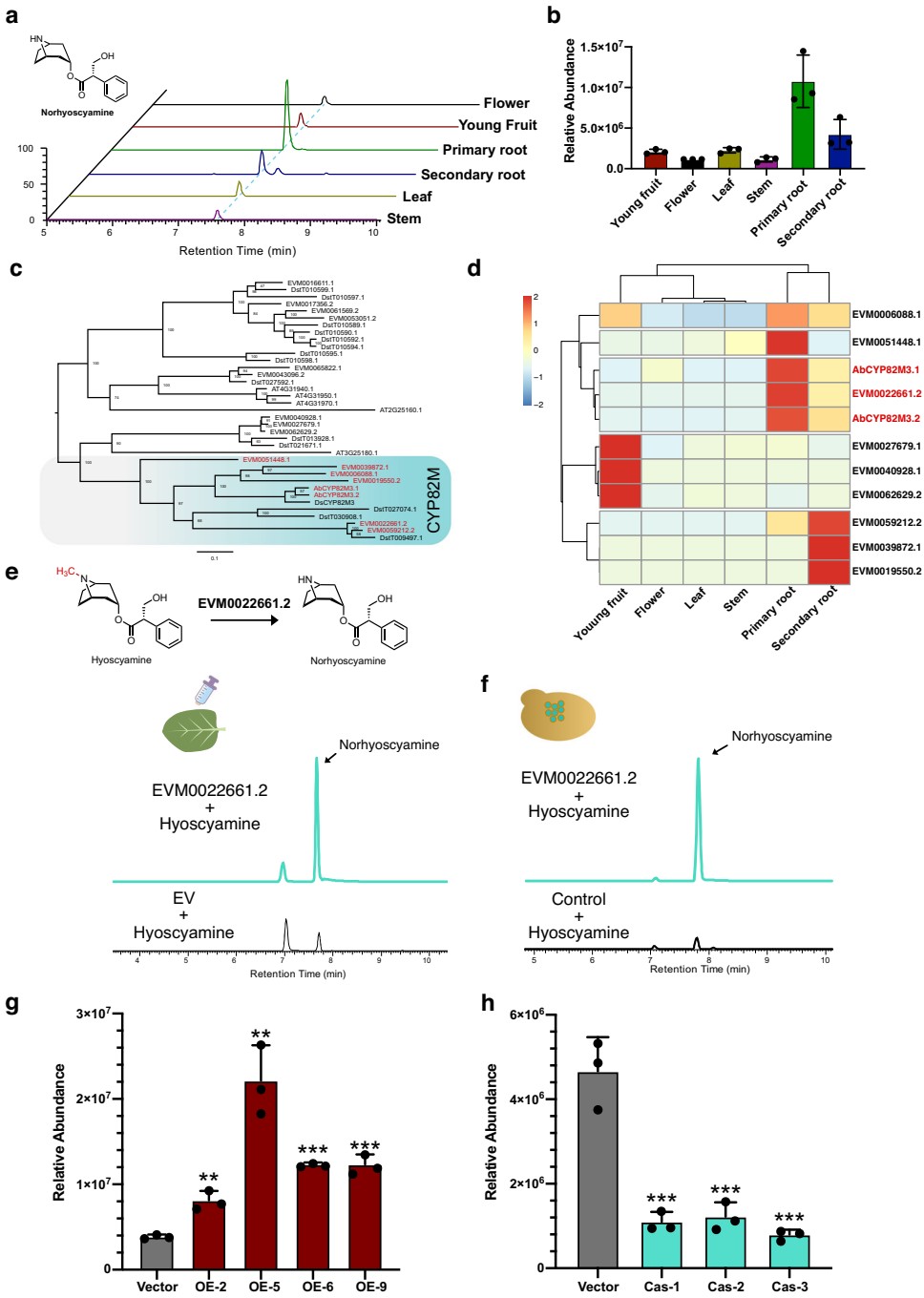

**Fig. 6 | Characterization of the *N*-demethylase of hyoscyamine in *A. belladonna*.** **a** Analysis of norhyoscyamine content in different tissues of *A. belladonna* by UPLC-MS. The light blue line represents the retention time of norhyoscyamine. **b** The relative abundance of norhyoscyamine in different tissues of *A. belladonna*. The data are presented as means values +/− s.d. (*n* = 3 biologically independent samples). **c** The maximum-likelihood tree of the CYP82 subfamily in *A. belladonna*, *D. stramonium* and *A. thaliana*. The CYP82M clade is highlighted. The bootstrap support value (*n* = 1000) is shown on the branch. **d** The coexpression analysis of CYP82M family members in different tissues of *A. belladonna*. The expression values are scaled as TPM. **e** Functional identification of the *N*-demethylase activity of EVM0022661.2 by tobacco transient expression assays. Extracted ion chromatograms showing the in vivo *N*-demethylase activity of EVM0022661.2 in tobacco leaves. Hyoscyamine was used as a substrate by infiltrating tobacco leaves. **f** Functional identification of the *N*-demethylase activity of EVM0022661.2 using

microsomes extracted from a yeast strain expressing EVM0022661.2; the untransformed yeast strain was used as a negative control. **g** The relative abundance of norhyoscyamine in the EVM0022661.2 overexpressed *A. belladonna* hairy roots. The data are presented as means values +/− s.d. (*n* = 3 biologically independent samples). ** represents significant difference from control line (Vector) analyzed by two-sided Student's *t*-test at the level of *P* < 0.01. *** represents significant difference from control line (Vector) analyzed by two-sided Student's *t*-test at the level of *P* < 0.001. **P* = 0.0030 (OE-2), **P* = 0.0016 (OE-5), ***P* = 0.0000 (OE-6), ***P* = 0.0003 (OE-9). **h** The relative abundance of norhyoscyamine in the EVM0022661.2 genome-edited *A. belladonna* hairy roots. The data are presented as means values +/− s.d. (*n* = 3 biologically independent samples). *** represents significant difference from control line (Vector) analyzed by two-sided Student's *t*-test at the level of *P* < 0.01. ***P* = 0.0018 (Cas-1), ***P* = 0.0024 (Cas-2), ***P* = 0.0012 (Cas-3). Source data underlying **b**, **d**, **g** and **h** are provided as a Source Data file.

*AbLS* can be found in the syntenic region of *Iochroma cyaneum* (Fig. 5d; Supplementary Figs. 34, 35). Moreover, the microsynteny analysis of *CYP80F1* provides more evidence. Similar to the *LS* gene, the genomic blocks of the *CYP80F1* gene likewise showed a high degree of synteny across Solanaceae, whereas only *DsCYP80F1* and *AbCYP80F1* were syntenic. Nevertheless, two exons of *CYP80F1* from *A. belladonna* can be found in hot pepper and tomato. All the evidence related to *LS* and *CYP80F1* indicates that the mTAs biosynthesis pathway might exist in the early diverging Solanaceae ancestor, probably before the *P. axillaris* speciation event, and gene loss led to the uneven distribution of mTAs in Solanaceae. Although evolutionary innovation by neofunctionalization or subfunctionalization through gene duplication has been reported in many species, a perspective of gene loss as a pervasive source of genetic change that causes phenotypic diversity has been proposed since the burst of high-quality genome sequencing (reviewed by Albalat et al.[48]). The pervasiveness of gene loss in most life forms suggests that gene loss would be a general force that affects all organisms. For instance, the biased loss of the wingless (*Wnt*) gene, a key regulator involved in animal cell fate determination during tissue differentiation, in different animal taxa contributes to the varying shapes of animals[49]. In Solanaceae, the loss of *AN2* leads to white flowers in *P. axillaris*, which might change its pollinators[50]. Similarly, the asymmetric loss of medium chain acyl sugar biosynthesis genes also determined the distribution of medium chain acyl sugar in Solanaceae[46].

## Methods

### Plant materials
*A. belladonna* and *D. stramonium* were grown in a greenhouse at 24 °C under a light period of 16-h light/8-h dark. The fresh leaves were collected to exact high-quality genomic DNA for genome sequencing. Fresh leaves of *Lycium chinense* were collected from Anning District, Gansu Province, for high-throughput full-length sequencing. For transcriptome sequencing, tissues of *A. belladonna* and *D. stramonium* were sampled, including leaf, stem, flower, fruit, primary roots, and secondary roots tissues. To validate the TAs and mTAs contents in *S. lycopersicum*, *S. tuberosum*, *S. melongena*, *C. annuum*, *L. chinense*, *N. tabacum*, and *P. hybrida*, we cultured those plants in our greenhouse at 24 °C under a light period of 16-h light/8-h dark. To perform transient expression assays in tobacco leaves, the *Nicotiana benthamiana* were grown in the greenhouse at 28 °C under a light period of 16-h light/8-h dark.

### Genome sequencing
For Nanopore sequencing of *A. belladonna*, high-quality genomic DNA was extracted from young leaves using the cetyltrimethylammonium bromide (CTAB) method and purified by a QIAGEN DNA purification kit. For each Nanopore library, high-quality genomic DNA fragments (>20 Kb) were selected using BluePippin and used to construct long-read libraries following the protocols for the ONT platform (https://nanoporetech.com). The libraries were sequenced using GRIDION X5 (v 9.4.1; Oxford Nanopore Technologies) with 22 nanopore flow cells and the SQK-LSK108 sequencing kit. Base-calling of the raw nanopore reads was performed using the Oxford Nanopore base caller GUPPY (v 3.2.2; Oxford Nanopore Technologies) with default parameters.

For circular consensus sequencing of *D. stramonium*, high-quality genomic DNA was isolated from young leaves of *D. stramonium* using the CTAB method. A 15 kb DNA SMRTbell library was constructed following the protocol for the PacBio Sequel2 platform and the circular consensus sequencing (CCS) was performed; these sequencing reads are known as highly accurate long reads, or HiFi reads.

For Illumina sequencing of *A. belladonna* and *D. stramonium*, genomic DNA was extracted from young leaves using the CTAB method and broken into random fragments. Short-read libraries of *A. belladonna* and *D. stramonium* were constructed according to Illumina's standard protocol, and paired-end reads (2 × 150 bp) were sequenced on an Illumina HiSeq X Ten platform.

To construct Hi-C libraries, young leaves of *A. belladonna* and *D. stramonium* were fixed in 1% formaldehyde for crosslinking. Cells were lysed using a Dounce homogenizer and digested using the *Hin*d III restriction enzyme. The DNA ends were filled and labeled with biotin and the filled-in *Hin*d III sites were ligated to form *Nhe* I sites. Complexes containing the biotin-labeled ligation products were purified and sheared, and the biotinylated Hi-C ligation products were pulled down and used to construct Illumina sequencing libraries[51].

### Genome size estimation
Genome size was estimated by *K*-mer frequency distribution analysis (genome Size=*K*-mer_num/Peak_depth)[52]. First, the short reads were filtered using fastp (v 0.19.4)[53] with default parameters. Subsequently, the *K*-mers were counted using Jellyfish (v 2.2.10)[54] with the parameter "-C -m 51 -s 10000000000 -t 50". The output file was then used as input for GenomeScope[55] to estimate the genome size with default parameters. The frequency distribution of 17-mers is based on the genome characteristics and in the light of the pattern of Poisson distribution.

### Genome assembly
To assemble the contigs of *A. belladonna*, the ONT reads were first corrected and trimmed using Nextdenovo (v2.0-beta.1) (parameters: read_cuoff = 1k, seed_cutoff = 28k, blocksize = 8 g) (https://github.com/Nextomics/NextDenovo.git). The corrected ONT reads were directly assembled using SMARTDENOVO (v 1.0.0) with default parameters. (https://omictools.com/smartdenovo-tool)[56]. The assembled contigs were polished four times by Pilon (v 1.18)[57] and NextPolish (v 1.0.5)[58] with default parameters, using 53.88 Gb (34×) of Illumina short reads, to yield high-quality contigs of *A. belladonna*.

To primarily assemble the genome of *D. stramonium*, a total of 828.66 Gb of raw PacBio subreads were filtered and corrected using the pbccs pipeline with default parameters (https://github.com/PacificBiosciences/ccs). The resulting CCS reads were subjected to hifiasm (v 0.14)[59] for de novo assembly with default parameters (https://github.com/chhylp123/hifiasm). We corrected the primary contigs with the Pilon (v 1.18)[57] program with default parameters using 145.27 Gb (79×) of Illumina paired-end reads. BWA (v 0.7.10-r789)[60] and SAMtools (v 1.9)[61] were used for read alignment and SAM/BAM format conversion.

### Chromosomal genome construction
Hi-C Pro (v 2.11.4)[62] was used to validate paired-end reads with default parameters. Draft contigs were clustered and ordered into chromosomes following intrachromosomal interactions estimated by LACHESIS (v 1.0.0)[63]. Validated paired-end reads were also used to calculate interchromosomal interactions to analyze chromosome territories. The contigs were further independently assembled into scaffolds using SLR (v 1.0.0, https://github.com/luojunwei/SLR)[64] and SALSA (v 2.2)[65] to validate the accuracy of the LACHESIS assembly with default parameters. To assess the completeness of *A. belladonna* and *D. stramonium*, we performed Benchmarking Universal Single-Copy Orthologs (BUSCO, v 5.3.2) analysis against embryophyta_odb10 (release Sep 2020) with default parameters[66].

### Annotation of genome sequences
To enable better parallel computation and accelerate the annotation of repetitive sequences, we used the scaffolds rather than the chromosome-level genome assembly. Tandem repeats and TEs were predicted separately. Tandem repeats were identified using Tandem Repeats Finder (v 4.0.9)[67], which was implemented with 'Match = 2, Mismatch = 7, Delta = 7, PM = 80, PI = 10, Minscore = 50 and MaxPeriod = 2000'. Then, for TE identification, a combination of homology-

based and de novo approaches were mainly used. We first used RepeatMasker (v 4.0.5)[68] (http://www.repeatmasker.org) with the Repbase TE library[69] and RepeatProteinMasker (v 1.36)[68] with the TE protein database to search for homologous repeat sequences in the genome with default parameters. Then, de novo-based identification was performed by RepeatModeler (v 1.0.9)[68] (http://www.repeatmasker.org/RepeatModeler.html) and LTR_FINDER (v 1.0.6)[70] with default parameters to predict the repeat element boundaries and family relationships from genome data. Finally, all repeat identification results from different software were integrated with redundancy elimination as the final repeat annotation results. We applied the same protocol to annotate the genome sequences of *A. belladonna* and *D. stramonium* for further comparison.

Three complementary methods incorporated in the MAKER pipeline (v 2.31.9)[71] were employed to predict the high-quality protein-coding genes: homology-based, de novo, and transcriptome-based predictions. In homology-based predictions, protein sequences of six species (*A. thaliana, S. tuberosum, S. lycopersicum, C. annuum, N. tabacum, S. pennellii,* and *S. melongena*) were downloaded from the Phytozome database (https://phytozome.jgi.doe.gov/pz/portal.html) and aligned to the repeat-masked genomes of two species by BLASTN with an E-value cutoff of 1e-5, and gene models were defined using GeMoMa (v 1.3.1)[72] with default parameters. The aligned sequence and candidate genomic regions were corrected and optimized by GeneWise (v 2.4.1)[73] with default parameters to further predict the exact structure of the protein-coding gene. For de novo prediction, 3000 full-length genes were randomly selected from the homology-based prediction results to train gene models of the two species. Four de novo prediction programs, including Augustus (v 3.2.1)[74], GlimmerHMM (v 3.0.4)[75], Genscan (v 1.1)[76], and SNAP (v 2006-07-28)[77], were utilized with *A. belladonna* and *D. stramonium* gene models for de novo prediction with default parameters. Genes with coding sequences <150 bp were discarded. For transcriptome-based predictions, we first performed end-trimming by SeqClean (http://www.tigr.org/tdb/tgi/software) for transcriptome assembly. Then, all transcript sequences from different tissues (flowers, leaves, stems, fruits, primary roots, and secondary roots) were aligned to the genome by the PASA pipeline (v 2.1.0)[78] with default parameters, and TransDecoder (v 3.0.0, https://github.com/TransDecoder) was used to produce the annotation file. All predictions of gene models from the above approaches were finally integrated using EVidenceModeler software (EVM; v 1.1.1)[79] with default parameters to generate consensus gene sets. The completeness of the gene sets was evaluated for both species with BUSCO (v 5.3.2)[80] against embryophyta_odb10.

## Noncoding RNAs (ncRNAs) and functional annotation

Noncoding RNAs and small RNAs were annotated by alignment to the Rfam and miRNA databases using INFERNAL (v 1.1.2)[81] and BLASTN with default parameters, respectively. The tRNA genes were identified using the tRNAscan-SE software (v 2.0)[82] with default parameters. rRNA fragments were predicted by alignment to rRNA sequences based on BLASTN analysis (E-value of 1e-10). Functional annotations of the predicted protein-coding genes were performed (E-value of 1e−5) against publicly available protein databases including KEGG[83], SwissProt[84], NR databases[85], and InterPro[86]. InterProScan (v 4.8) and HMMER (v 3.3)[87] were used to query the InterPro and Pfam[88] databases to identify the protein domain, respectively. Then, GO terms were assigned by the Blast2GO pipeline (v 3.1.3)[89] with default parameters. GO enrichment and KEGG pathway analysis were performed using an online platform, OmicShare (https://www.omicshare.com/).

## Identification and classification of TEs

Long terminal repeat retrotransposons (LTR-RTs) were initially identified using LTRFinder (v 1.02)[70] and LTRharvest (v 1.5.10)[90]. LTR_retriever (v 3.0)[91] was then used to filter out false LTR-RTs using

structural and sequence features of target site duplications, terminal motifs, and LTR-RT Pfam domains with default parameters. Finally, LTR-RTs were annotated by RepeatMasker using the constructed nonredundant LTR-RT library and intact LTR insertion time provided by LTR_retriever. The identification parameters were as follows: for LTR-harvest, overlaps best -seed 20 -minlenltr 100 -maxlenltr 2000 -mindistltr 3000 -maxdistltr 25000 -similar 85 -mintsd 4 -maxtsd 20 -motif tgca -motifmis 1 -vic 60 -xdrop 5 -mat 2 -mis -2 -ins -3 -del -3, and for LTR-finder: -D 15000 -d 1000 -L 7000 -l 100 -p 20 -C -M 0.9. The two datasets were integrated to remove false positives using the LTR-retriever package. The insertion time was estimated using the formula

$$T = Ks/2r \tag{1}$$

where $Ks$ is the divergence rate and $r$ ($3.48 \times 10^{-9}$) is the substitution rate.

## Genome evolution analysis

To clarify the phylogenetic relationships of *A. belladonna*, *D. stramonium*, and *Lycium chinense*, we selected 10 additional species for analyses of WGD events. The homologous groups among all 13 species were identified using the OrthoMCL (v 2.0.9) method[92]. The homologous clusters were obtained using the Markov graph clustering (MCL; v 14.137) algorithm, through an all-vs-all sequence similarity search by BLASTP (v 2.2.29) with an E-value cutoff of 1e−3. Then we extracted single-copy homologous genes from OrthoMCL results. Protein sequences were aligned by MAFFT (v 7.407)[93] and Gblocks (v 0.91b)[94] with default parameters to extract conserved sites of multiple sequence alignment results and construct a phylogenetic tree by RAxML (v 8.1.13)[95] with the GTRGAMMA model for amino acid sequences and *Oryza sativa* as the outgroup. Then, 1000 bootstrap analyses were performed to test the robustness of each branch. To estimate the divergence time, we extracted the 4-fold sites from the 229 homolog pairs and estimated the divergence time using MCMCTree in the PAML (v 4.9e) package with the "relaxed-clock (clock = 2)" model and "F84" model[96]. The divergence times of *Vitis vinifera* and *Oryza sativa* (mean: 160.0 Mya, std dev: 4.0) obtained from the TimeTree database (http://www.timetree.org) were applied to calibrate the divergence times. CAFE (v 3.1) was used to identify expansions and contractions of gene families with default parameters following divergence predicted by the phylogenetic tree with a probabilistic graphical model[97]. A conditional *P* value was calculated for each gene family, and families with conditional *P* values < 0.05 were considered to have had a significantly accelerated rate of expansion or contraction.

## Genome synteny and whole-genome duplication (WGD) event analysis

The distribution of synonymous substitutions per site (*Ks*) within paralogs was used to examine the most recent WGD event in *A. belladonna* and *D. stramonium*. For intergenomic comparison, we compared the *A. belladonna* and *D. stramonium* genomes with those of five other species (eggplant, tobacco, tomato, grape, and pepper) or with themselves. The homologs between these species were identified using MCScanX (v 11-13-2012)[98]. Subsequently, the *Ks* substitution rates of the gene pairs in syntenic blocks were calculated. Finally, we illustrated the *Ks* distribution and generated dot plots of homologous blocks using MCScanX. *Ks* values between colinear genes were estimated using the CodeML approach as implemented in the PAML package[96]. Finally, we illustrated the *Ks* distribution and created dot plots of homologous blocks using MCScanX. The CIRCOS (v 0.69.6) software was used to visualize gene density, GC content, repeat content, and gene synteny on individual pseudochromosomes[99].

### RNA-seq sequencing and gene coexpression analysis

Total RNA was extracted using a RNeasy Plus Mini Kit (Qiagen). Then mRNA isolation, fragmentation, and purification were performed using a TruSeq RNA Library Prep Kit v.2 (Illumina, San Diego, CA, USA). The libraries were sequenced using the Illumina NextSeq 500 platform. Raw RNA-seq data (Supplementary Table 25) were trimmed using TRIMMOMATIC (v 0.39)[100]. Cleaned reads were mapped to genome assembly guided by gene annotation models using HISAT2 (v 2.0.5)[101] with default parameters. The read counts and transcripts per million (TPM) values for each gene was performed by StringTie (v 1.3.3b)[102]. After then the read counts and TPM values were filtered using the sva R package (v 3.11) to decrease batch effects and hidden variables[102]. Differentially expressed genes (DEGs) were detected using DESeq2 (v 1.27.12)[103] based on absolute log2 transformed fold-change values >2 and a *P* value of 0.05 after applying the Benjamini–Hochberg correction[104].

To identify relationships between differentially expressed genes, we performed weighted gene coexpression analysis with the R package WGCNA (v.1.69)[105]. All R packages were run in version 3.1 of R. The expression data were prefiltered using the built-in quality control function. A signed coexpression network was constructed using a soft-thresholding power of 8 and default parameters. The only exception was the mergeCutHeight parameter, controlling the minimum distance between coexpression clusters, which was set to 0.25. Finally, we obtained 30 clusters for these genes. Then we used Cytoscape (v 3.5.1)[106] to display the network. Network statistics were calculated using NetworkAnalyzer in Cytoscape.

### Full-length RNA sequencing and analysis of *L. chinense*

The total RNA of young *L. chinense* leaves was extracted using the RNeasy Plus Mini Kit (Qiagen). RNA quality was analyzed using the Plant RNA Nano assay of a 2100 Bioanalyzer (Agilent). PacBio library preparation and sequencing were performed according to the Iso-Seq protocol by Pacific Biosciences. Sequencing was performed on the PacBio RSII System. PacBio ROIs were processed using PacBio's Iso-seq ToFU tool[107] and PacBio SMRT Analysis (v 5.1) with default settings. Then, we classified the reads into full-length and non-full-length reads by ToFU pbtranscript classify with default parameters, and the full-length reads were clustered and error corrected using ToFU pbtranscript cluster with default settings. The full-length sequences generated by ToFU were then clustered by CD-HIT (v 4.8.1)[108] to remove redundancy furthermore. Subsequently, putative coding regions were predicted using TransDecoder.

### Identification of genes involved in mTAs biosynthesis

According to our previous studies[74], we downloaded first-generation sequencing data of the key enzymes in the TAs biosynthetic pathway from the National Center for Biotechnology Information (NCBI) database. We annotated the key enzyme genes by BLAST (v 2.2.28) and aligned them with the Pfam database by HMMER (v 3.3)[87] with default parameters in *A. belladonna*, *D. stramonium*, and eight other Solanaceae species.

### *MPO* RNA interference transgenic hairy root establishment and tropane alkaloid content analysis

The 521-bp fragment of *AbMPO1* (*EVM0017027.2*) and *AbMPO2* (*EVM0068072.1*) was amplified and inserted into the RNA interference vector pHannibal, generating AbMPO1-RNAi and AbMPO2-RNAi cassettes. Thereafter, the cassettes were inserted into the plant expression vector pBIN19 and then transferred into *Agrobacterium tumefaciens* strain C58C1, which was used to infect leaf explants obtained from 4-week-old *A. belladonna* plants to initiate root cultures on MS medium[6]. Five independent positive transformants were selected, and the relative expression levels of *AbMPO1* and *AbMPO2* were measured by qRT-PCR (Supplementary Fig. 11). The contents of

tropane alkaloids in hairy root cultures were analyzed by high-performance liquid chromatography (HPLC). The HPLC system was a Shimadzu LC- 20 A instrument (Shimadzu Corp., Kyoto, Japan), and the detector was the photo-diode array. The detecting wavelength was 226 nm. The mobile phase consisted of $CH_3CN$: 20 mM $NH_4OAc$ (11:89, v/v), and the $NH_4OAc$ solution included 0.1% EtOAc (pH 4.0). To exact the alkaloids, 200 mg fine powdered dry hairy roots were immersed in the exaction buffer (chloroform/methanol/17% ammonia, 15:5:1, v/v/v) over night, and then the filtered exaction buffer was evaporated at 40 °C. The dried samples were dissolved in 5 ml of chloroform. Subsequently, 2 mL $H_2SO_4$ (0.5 M) was added to extract the alkaloid into the aqueous phase, which was transferred into a 10 mL EP tube. 1.5 mL of ammonia (17%) and 2 mL chloroform was added in and then the chloroform phase was transferred into a beaker and placed in an oven at 40 °C to evaporate the chloroform. The dryness was dissolved in 1 mL methanol. After centrifugation at $13,000 \times g$ for 5 min, the supernatant was used for HPLC analysis. All the primers used in the functional characterization of MPO are listed in Supplementary Table 26.

### Quantification of tropane alkaloids by UPLC-MS

To quantify tropane alkaloids by ultra-high-performance liquid chromatography coupled mass spectrometry (UPLC-MS), the plant samples were lyophilized and ground into a fine powder. One milliliter of extraction buffer containing 20% methanol, and 0.1% formic acid was added to 25 mg of each sample powder. After centrifugation at $13,000 \times g$ for 5 min, the supernatant was used for UPLC-MS analysis.

We detected the littorine, hyoscyamine, scopolamine, and norhyoscyamine by a Thermo Scientific UltiMate 3000 UPLC system with a Thermo Scientific Q Exactive Orbitrap LC-MS instrument. A 3-μL volume of each extract was analyzed using a Hypersil GOLD C18 column (2.1 mm × 100 mm, 1.9 μm) obtained from Thermo Scientific (Pittsburgh, PA, USA). Measurements of littorine, hyoscyamine, scopolamine, and norhyoscyamine were taken using electron spray ionization (ESI) in positive ion mode and full MS mode. The instrument parameters were as follows: source voltage of 3.0 kV; capillary temperature of 350 °C; S-lens RF level of 50; and auxiliary gas heater temperature of 350 °C. To detect phenyllactate, the same UPLC-MS system and a Symmetry C18 Column with the mobile phases added 0.02% formic acid and instrument parameters were used, but a 3 μL volume of each extract was separated and detected using ESI in negative ion mode and full MS mode. To detect tropinone and tropine, the same UPLC-MS system with the same instrument parameters was used, but a 3-μL volume of each extract was separated by an ACQUITY UPLC BEH HILIC column (2.1 mm × 100 mm, 1.7 μm) obtained from Waters (Milford, MA, USA), and detected using ESI in positive ion mode and full MS mode.

### PPAR enzyme activity assay

To perform the PPAR activity assays, the coding sequences of *PPARs* were cloned from the corresponding species and inserted into the protein expression vector pET28a. Recombinant His-tagged PPARs were purified using HisPur Ni-NTA resin. The purified PPARs (20 μg) were incubated with 2 mM phenylpyruvic acid in reaction buffer (500 μL) containing 2 mM NADPH in 50 mM potassium phosphate (pH 8.0) at 30 °C for 2 h, while boiled enzyme was used as a negative control. The products were detected by UPLC-MS as described in the above section on quantification of tropane alkaloids by UPLC-MS. The enzymatic kinetics of PPARs were determined for 1 h at pH 8.0. The 250 μL reaction buffer containing 20 μg of PPAR protein, 2 mM NADPH +, and phenylpyruvic acid (0.1–14 mM) for the determination of $K_m$ values in 0.1 M Tris-HCl buffer (pH 8.0). The kinetic constants were calculated with a nonlinear regression of the Michaelis–Menten equation using OriginPro (v 8.0, OriginLab). All assays were repeated three times, and mean values with standard deviations are reported. All

the primers used for cloning *PPAR*s and constructing protein expression plasmids are listed in Supplementary Table 27.

## LS enzyme activity assay

To evaluate the activity of the LS enzyme, *AbUGT1* from *A. belladonna* and *LS* from different plants were transiently expressed in *N. benthamiana* leaves. First, the full-length CDSs of *AbUGT1*, *DsLS*, *AbLS*, and *LcLS* were cloned into the transient expression vector pEAQ-HT using the restriction enzymes *Age*I and *Xho*I to generate pEAQ-AbUGT1, pEAQ-DsLS, pEAQ-AbLS, and pEAQ-LcLS, respectively. Subsequently, they were independently transformed into *Agrobacterium* GV3101. Ten milliliters of overnight cultured engineered *Agrobacterium* were centrifuged, and the pellets were resuspended in liquid MS medium containing 10 mM MES, 10 mM MgCl2, and 150 mM acetosyringone to an OD600 of 0.5–0.6. Tobacco leaves were infiltrated with resuspension by using a 1 mL needleless syringe. Four days later, tobacco leaves were infiltrated with substrate solution containing 1 mM tropine and 1 mM phenyllactate. Finally, tobacco leaves infiltrated with substrates were harvested for metabolite analysis after 24 h. Each group had six biological replicates. The littorine was quantified by UPLC-MS as described in the above section on quantification of tropane alkaloids by UPLC-MS. The primers used for cloning *LS*s and constructing overexpression plasmids are listed in Supplementary Table 28.

## In vitro TRI enzyme activity assay

To characterize the activity of TRI from *A. belladonna*, *D. stramonium*, *C. annuum*, *S. lycopersicum* and *S. tuberosum*, we cloned the coding sequence of *TRI*s from the cDNA of those plants and inserted it into the protein expression vector pET28a. Recombinant His-tagged TRIs were purified using HisPur Ni-NTA resin. To detect the products of TRIs, His-tagged TRIs (5 μg) were incubated with tropinone (100 mM) in the presence of NADPH (200 μM) for 2 h at 30 °C. Subsequently, the reaction products were extracted and analyzed by UPLC-MS as described in the above section on quantification of tropane alkaloids by UPLC-MS. We detected the activity of TRI by measuring NADPH + H$^+$ consumption, substituted with the direct product tropine, using a spectrophotometer (U3010, HITACHI, Japan) at 340 nm and 30 °C. Then, 1 mL samples containing 20 μg of TRI protein, 200 μM NADPH + H$^+$, and tropinone (0.01–30 mM for the determination of $K_m$ values, as the concentration range of each substrate needed to be adjusted according to the results of enzymatic assays), and 0.1 M potassium phosphate to yield pH 6.4. Data were collected during the initial linear phase of the enzyme reaction to calculate the kinetic parameters. The kinetic constants were calculated with a nonlinear regression of the Michaelis–Menten equation using OriginPro (v 8.0, OriginLab). All assays were repeated three times, and mean values with standard deviations are reported. The primers used for *TRI*s cloning and constructing protein expression plasmids are listed in Supplementary Table 29.

## CYP82M3 enzyme activity assay

To characterize the activity of CYP82M3, the CDSs of *CYP82M3* from *C. annuum* and *S. lycopersicum* were cloned and inserted into the transient expression vector pEAQ-HT, generating pEAQ-CaCYP82M3 and pEAQ-SlCYP82M3. The CDSs of *AbPMT* and *AbPYKS* were cloned from *A. belladonna* and inserted into pEAQ-HT as well to generate pEAQ-AbPMT and pEAQ-AbPYKS, which provided the substrates of CYP82M3. Subsequently, those vectors were transferred into *Agrobacterium* GV3101, and then the resulting *Agrobacterium* were infiltrated into tobacco leaves. Four days later, infiltrated tobacco leaves were harvested to detect tropinone by UPLC-MS as described in the above section on quantification of tropane alkaloids by UPLC-MS. The primers used for *CYP82M3* cloning and constructing overexpression plasmids are listed in Supplementary Table 30.

## N-demethylase activity assay

To evaluate the *N*-demethylase activity of EVM0022661.2, methods similar to those described above for LS enzyme activity assays were employed. The CDS of *EVM0022661.2* was amplified from *A. belladonna* and inserted into the pEAQ-HT vector. Subsequently, EVM0022661.2 was transiently expressed in the tobacco leaves and hyoscyamine was infiltrated and used as a substrate after four days. To perform the in vitro *N*-demethylase activity, the CDS of *EVM0022661.2* was inserted into the yeast expression vector pYES2 and transformed into the yeast strain WAT11. An untransformed yeast strain was used as the negative control. The yeast cells were collected by centrifugation and the generated pellets were resuspended twice in TEK (50 mM Tris-HCl, pH 7.4, 1 mM EDTA, 0.1 M KCl) and TESB (50 mM Tris-HCl, pH 7.4, 1 mM EDTA, 0.6 M sorbitol) buffers. Subsequently, the pellets were broken up by a cryogenic homogenizer and the precipitated microsome pellets were collected by further centrifugation. Consequently, 0.5 mg of resuspended microsomal protein and 100 μM hyoscyamine were included with the reaction buffer (500 μL) at 30 °C for 2 h. The detection method of norhyoscyamine was the same as that of hyoscyamine, as described in the above section on quantification of tropane alkaloids by UPLC-MS.

## *EVM0022661.2* overexpressed and genome-edited transgenic hairy root establishment

To overexpress *EVM0022661.2*, the CDS of *EVM0022661.2* was cloned and inserted into the plasmid pBI121, generating plasmid EVM0022661.2-pBI121. To edit *EVM0022661.2* in the hairy root of *A. belladonna* via CRISPR technology, a 20-bp fragment of the *EVM0022661.2* was used as the targeting sequence for genome editing (Supplementary Fig. 49). Subsequently, the synthesized primers containing targeting sequence was inserted into the p1300-Cas9N vector to generate the genome editing plasmid Cas9N-EVM0022661.2. The constructed plasmids, EVM0022661.2-pBI121 and Cas9N-EVM0022661.2, were then transferred into *Agrobacterium tumefaciens* strain C58C1. The hairy root of *A. belladonna* was established and cultured as described in the above section on *MPO* RNA interference transgenic hairy root establishment and tropane alkaloid content analysis. The primers used for *EVM0022661.2* cloning and constructing plasmids for overexpression and genome edition are listed in Supplementary Table 31.

## Reporting summary

Further information on research design is available in the Nature Portfolio Reporting Summary linked to this article.

## Data availability

The genomic sequencing and transcriptome data generated in this study have been deposited in the NCBI Sequence Read Archive (SRA) database under accession PRJNA766188 for *Atropa belladonna* and PRJNA765895 for *Datura stramonium*. The full-length transcriptome data of *Lycium chinense* have been deposited in the NCBI Sequence Read Archive (SRA) database under accession PRJNA769498. The genome assembly and annotation of *A. belladonna* and *D. stramonium* are also available at Genome Warehouse of the National Genomics Data Center under accession GWHBOWM00000000 [https://ngdc.cncb.ac.cn/bioproject/browse/PRJCA012583] and GWHBOZL00000000 [https://ngdc.cncb.ac.cn/bioproject/browse/PRJCA012583], respectively. Source data are provided with this paper.

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

## Acknowledgements

This work was supported by the National Natural Science Foundation of China (grant number: U1902212, Z.L.; 31770335, Z.L.; 81803660, F.Z.; and 32270276, F.Q.); the National Transgenic Major Project of China (grant number: 2019ZX08010-004, Z.L.); the Fourth National Survey of Traditional Chinese Medicine Resources, Chinese or Tibet Medicinal Resources Investigation in Tibet Autonomous Region (State Administration of Chinese Traditional Medicine, grant number: 20191217-540124, X.L. and Z.L.; 20191223-540126, X.L. and Z.L.; 20200501-542329, X.L. and Z.L.); Key project at central government level: The ability establishment of sustainable use for valuable Chinese medicine resources (2060302, F.Q.).

## Author contributions

Z.L. led the project planning. F.Z., F.Q., J.Z., Z.X., T.Z., Y.G., F.S., S.W., X.S., and W.W. carried out experiments. F.Z., Z.X., J.Z., and Z.X. performed the bioinformatic analysis. Z.L., F.Z., C.Y., L.Z., X.L., M.C., and J.Z. interpreted the data and participated in discussion. F.Z. wrote the paper. Z.L., Y.T., and M.C. revised the paper.

## Competing interests

The authors declare no competing interests.
