## [Peer Review File · Nature Communications]

The evolution of tropane alkaloid biosynthesis in the SolanaceaeReviewers' Comments:

Reviewer #1:

Remarks to the Author:

The manuscript of Zang and co-workers describes the sequencing and analyses of chromosome-scale genomes of the two plant species of interest, namely *Atropa belladonna* and *Datura stramonium*, which both produce valuable tropane alkaloids (TAs) such as hyoscyamine and scopolamine. By performing comparative analyses, homology searches, microsynteny analyses and functional studies, the authors provide a detailed and complete description of the evolution mechanisms leading to the acquisition of the TA biosynthesis capacity with a conserved gene cluster, gene duplication and losses. The authors also show that *A. belladonna* and *S. Stramonium* synthesizes complex TAs through a conserved biosynthetic pathway albeit being two distantly related species. Lastly, they identified a P450 catalyzing a missing step in TA synthesis relying on hyoscyamine demethylation. Experiments have been well conducted, the results are convincing, and the manuscript is well written. Comments and Concerns are listed below:

- Why has 2 different long-read DNaseq been used?
- All versions of software and parameters should be mentioned.
- Complete BUSCO scores (%CS, %CD, %F, %M) should be reported
- L226: "The genome size of *A. belladonna* and *D. stramonium* were significantly large..." What significance test has been used ?
- Reference to previous studies is missing 369 ("which were documented in many Solanaceae species").
- "Genome sequencing" and "Genome Size Estimation and Assembly" should be reorganized and clarified.
- It is unclear to which species belong the Illumina DNaseq sequencing part (L608-611)
- RNA sequencing part for *L. chinense* is missing in Material and Method section
- BUSCO results for annotated genes are not shown
- Data for *L. chinense* should also be made available on NCBI and figshare.
- The authors performed a very interesting and complete microsynteny analysis at the whole TA biosynthetic pathway scale to provide information of the evolution of the pathway. However, the whole organization of the pathway should be described in more details at the chromosome level. What about the respective position of each TA gene compared to other TA genes? Are there any true clusters besides the tropane cluster? Indicating TA genes on figure 1 or similar would be nice to see distribution
- What about the genes surrounding TA genes? Are there any interesting features? Can they represent putative TA genes?
- Based on their test of activity in planta, the authors indicated that DsLS displays a 16-fold higher activity compared to AbLS. Is this result supported by analysis of similar amount of both protein accumulation in the leaf extracts used for assays? (checked by Western blot for instance)?
- The authors identified a P450 catalyzing N-demethylation of hyoscyamine. Albeit there is no doubt on the tests of activity they performed in both benth and yeast, one may imagine that this N-demethylase activity is not the primary of the enzyme (that could work with a higher efficiency on a distinct substrate). Have the authors considered the possibility to perform complementary analyses in planta by gene silencing for instance? albeit not mandatory, this would provide definitive evidences of the function of this enzyme.

Figures and tables:

- Fig.3A species pictograms are not easy to read and should be replaced by species abbreviation to ease reading.
- Please homogenize the figures S1, S2, S3
- Showing all species on Fig. S4 would have been interesting. You could try UpSet plot from ComplexHeatmap package
- X labels are missing on Fig. S6, Fig.S10B
- Please provide a legend for Fig.S41, S42b, S43
- Raw Illumina short-reads information is missing for *A. belladonna*. Similar information for all RNAseq

data would be needed

Other minor concerns:

L49: angiosperm should be written Angiosperm

L60: eudicots ◊ Eudicots

L316-317 is unclear and should be rephrased.

L629: "the full length transcriptomes were assembled" seems odd in a Genome assembly section

Reviewer #2:

Remarks to the Author:

The authors report the chromosome scale genome assemblies of the tropane alkaloid producing species *Atropa belladonna* and *Datura stramonium*. The authors utilize these assemblies to investigate the evolution of tropane alkaloid biosynthesis within the Solanaceae family. Not all members of the Solanaceae synthesize the medicinal tropane alkaloids reported in this manuscript, which creates the potential to investigate the evolutionary trajectory of these metabolites. The topic is of broad interest to researchers interested in plant specialized metabolism. The genomic resources will also be useful for biologists interested in comparative biology of the Solanaceae family. However, there are a number of concerns throughout the manuscript relating to data quality, interpretation, presentation, and availability that need to be addressed prior to the manuscript being suitable for publication.

Major Concerns

1. As presented, the WGCNA data presented in Figure S40 and S41 is uninformative. At a minimum, a supplemental dataset file (Excel sheet) should be presented that provides gene identifier information and annotation for the genes in each module. In the current form, the data are not useful for the reader.
2. The resolution of many of the supplemental figures is extremely poor, especially the synteny figures (S26 through S38 and S42). The text is too small and does not allow for the reading of the gene identifiers. When the reader zooms in the images are blurred. In addition, in the case of all of the synteny figures, reference should be made to the corresponding chromosome numbers that are referenced for each species so that the reader can clearly follow the syntenic regions being discussed.
3. There is a general lack of detail and information provided for most of the supplemental figures and supplemental tables. In many cases, legends are inadequate and do not provide the reader with necessary information to interpret the figure or the data.
4. Figure S1 is confusing. The shape of the curves for the two species is very different and the y-axes of each graph differs. Why is there this difference. Information should be provided that allows the reader to understand how the genome size of each species can be inferred from this k-mer analysis.
5. Figure S2b: In the contact map for *D. stramonium*, there appears to be small regions between LG4 & LG5 and LG9 & LG10 that are not necessarily part of either linkage group and show signal with other regions of the genome. Can the authors provide an explanation for these data?
6. Figure S3: There appears to be two GC peaks for the *D. stramonium* data. Why is this? Could it be associated with the transposable element content? Some of the graph axes do not have labels.
7. Figure 4 and associated manuscript text regarding the evolution of tropinone reductases: The authors should consider in their interpretation of their data that the tomato and potato TRI enzymes may actually have undergone neofunctionalization and that tropinone may not be their preferred substrate in planta. These genes have been retained in these species and may now have distinct biochemical activities that are yet undefined.
8. Figure S43: The authors should provide information regarding the sources of the gene expression data they are using for tomato, potato, and pepper.
9. The *D. stramonium* contig number provided in Table 1 does not match the number provided in Table S2.
10. The assembled genome size for *A. belladonna* provided in Table 1 does not match the figure provided in Table S2 and neither of these numbers match the assembled genome size provided on line

146 of the manuscript file. The authors should check these data.

11. In Table S4 the bulk of the *D. stramonium* genome assembly appears to be located in 85 contigs. However, depending on whether the contig number in Table 1 or Table S2 is correct, it means there are approximately 1000 contigs totaling close to 50 Mb that are not in the assembly. Can the authors provide some insight into this discrepancy.

12. Table S5 and S6: No footnotes or explanations are provided for these tables so the data is difficult to interpret. However, it is extremely unusual in the case of the de novo and the homology based annotations that many of the parameters reported have identical values in each table for the two different species. For example, it seems highly unlikely that the average intron length predicted by AUGUSTUS for both species be exactly the same value. The authors are asked to thoroughly check the data in these tables. Furthermore, the average gene length as predicted by Genscan seems particularly high for both species.

13. The data provided in Figure 3C may not be accurate as it relates to the evolution of the PPAR enzyme. For example, the authors have performed end-point assays containing very large amounts of recombinant protein (20 micrograms) and 2 millimolar phenylpyruvic acid. PPAR belongs to a family of enzymes that are catalytically promiscuous and it is possible that the authors are simply forcing a biochemical reaction to occur by utilizing large amounts of enzyme and substrate. The authors should be careful about drawing evolutionary conclusions from such experiments and would be better advised to perform kinetic analyses of these recombinant enzymes as they have completed for the TRI enzymes from different Solanaceae species. In addition, the specific gene IDs should be provided for each PPAR enzyme that was characterized.

14. The data describing the characterization of the hyoscyamine N-demethylase would be strengthened if RNAi or VIGS data was provided to support the enzyme activity data.

15. No table is provided that reports all of the primers that were utilized for gene cloning and construct assembly.

16. The manuscript would benefit from English language editing.

Minor comments:

1. Figure S10 b through e: The units on the x-axes for the graphs are not defined.

2. Figure S19: The gene name is tropinone reductase I. Not phenyllactate tropine forming reductase.

3. Figure S39: Gene name should be UGT1.

4. Table S10: Note footnotes provided. The abbreviation TRF is not defined.

5. Manuscript text lines 67 -68: The statement that not all tropanes have physiological activity is not entirely accurate. It's correct that some tropanes have very strong pharmaceutical activities. However, there are many tropanes where their bioactivities are simply not studied and the physiological roles are therefore not defined. The text should be edited accordingly.

6. Lines 82 and 83: UGT1 does not catalyze the formation of phenyllactate UDP glucose. It catalyzes the formation of phenyllactoyl glucose. UDP is a co-product of the reaction.

7. Line 270: please define HPD.

8. Figure 3D should read PPAR not PPR.

9. Line 768: I believe the vector name is pHannibal

Reviewer #3:

Remarks to the Author:

The manuscript entitled „The genomes of *Atropa belladonna* and *Datura stramonium* reveal the evolution of tropane alkaloid biosynthesis in the Solanaceae “ by Zhang and colleagues uses the newly assembled genomes of *Atropa belladonna* and *Datura stramonium* to shed light on the tropane alkaloid biosynthesis.

In total the authors have produced a solid manuscript using state of the art methods.

The main interesting findings were

1) chromosome level assemblies of *A. belladonna* and *D. stramonium*

- 2) Using hairy root transformation to show that an *A. belladonna* root isoform of MPO is likely the active isoform in roots
- 3) Showing that the duplication of TRI potentially allowed for more flexibility as this allowed a more root specific isoform as well as an enzyme better adapted to TRI activity. Surprisingly, also the authors could show that tomato TRI is active but at a very low level.
- 4) Revealing information about LS genes

Novelty

This report shows interesting information about the phylogeny and potentially evolution of tropane biosynthesis.

Comments

- 1) Since there are two isoforms of abMPO and one is root expressed wouldn't a root specific (i.e. hairy culture) downregulation of the less root expressed isoform abMPO2 be expected to have no major outcome? I would thus tone down the statement about MPO responsibility and/or also indicate that tropanes are usually synthesized in the roots.
- 2) Based on Figure 4 would the presence of multiple TRI genes in *Capsicum* and *Datura* not potentially be also explicable by a duplication of TRI that occurred in the Solanoideae [Särkinen 2013: comprising major Atropina, Solaneae, Capsiceae and Physaleae + small like Datureae] and lost in Solaneae (or the *Solanum* genus)? One could check easily in the *Physalis* genome even though a larger sampling would be interesting anyway.
- 3) It is good that the raw reads are publicly available in NCBI. However, the assembly with at least gene annotations must be made available in an INSDC database or a China CNCB resource and not on figshare to allow follow up studies
- 4) It would have been nice to speculate on the role of tomato TRI and to investigate which potential regions in the enzyme could lead to the difference in activity.
- 5) when describing genome sizes say e.g. tomato and tobacco as there are maybe larger genomes in the *Cyphomandra* genus than tobacco - just rewording is enough
- 6) line 225 gypsy expansion give citation or qualify statement
- 7) line 244 I wouldn't call the *Atropa* WGT species specific as close relatives were not investigated--- just wording

Reviewer #1 (Remarks to the Author):

The manuscript of Zhang and co-workers describes the sequencing and analyses of chromosome-scale genomes of the two plant species of interest, namely *Atropa belladonna* and *Datura stramonium*, which both produce valuable tropane alkaloids (TAs) such as hyoscyamine and scopolamine. By performing comparative analyses, homology searches, microsynteny analyses and functional studies, the authors provide a detailed and complete description of the evolution mechanisms leading to the acquisition of the TA biosynthesis capacity with a conserved gene cluster, gene duplication and losses. The authors also show that *A. belladonna* and *S. Stramonium* synthesizes complex TAs through a conserved biosynthetic pathway albeit being two distantly related species. Lastly, they identified a P450 catalyzing a missing step in TA synthesis relying on hyoscyamine demethylation. Experiments have been well conducted, the results are convincing, and the manuscript is well written. Comments and Concerns are listed

Response:

Thank you very much for these supportive comments on our work. We really appreciate all your valuable suggestions on our MS. We present point-to-point responses below.

Comment 1: Why has 2 different long-read DNAseq been used?

Response 1: Many thanks indeed for your kind and professional comments. We first sequenced the genome of *Atropa belladonna* in 2018. When we began to sequence the genome of *Datura stramonium* in 2021, the HiFi technology has been well established. The highly accurate long reads from HiFi technology could be helpful to improve the genome assembly of *D. stramonium*, whose genome is larger and more complex than *A. belladonna*. Therefore, we employed the HiFi technology to sequence the genome of *D. stramonium*. According to the BUSCO assessment and contig N50 values, the quality of the *D. stramonium* genome is comparable, if not better, to that of *A. belladonna*. Here, we believe that the completeness of *D. stramonium* and *A. belladonna* are of great quality to be reference-level.

Comment 2- All versions of software and parameters should be mentioned.

Response 2: Thanks for your kind recommendation. We have added versions and parameters of softwares we used in the revised manuscript (Lines 634-819).

Comment 3: Complete BUSCO scores (%CS, %CD, %F, %M) should be reported

Response 3: We have added BUSCO scores in Supplementary Tables 7 and 8 in the revised manuscript. And we also added the BUSCO analysis results of annotated genes as per your kind comments in Supplementary tables 7 and 8.

Comment 4: L226: “The genome size of *A. belladonna* and *D. stramonium* were significantly large...” What significance test has been used?

Response 4: We just simply compared the genome sizes of *A. belladonna* and *D. stramonium* (~1.65 Gb and ~1.80 Gb, respectively) with those of tomato and potato (~900 Mb and ~844 Mb, respectively). Thus, we have deleted the word “significantly” in the revised manuscript to make the manuscript more precise and accurate (Line 228).

Comment 5: Reference to previous studies is missing 369 (“which were documented in many Solanaceae species”).

Response 5: Thanks. We have added the references in the revised manuscript as follows (Line 372):

3. Griffin, W. J. & Lin, G. D. Chemotaxonomy and geographical distribution of tropane alkaloids. *Phytochemistry* **53**, 623–637 (2000).
4. Huang, J. P. et al. Tropane alkaloid biosynthesis: A centennial review. *Nat. Prod. Rep.* **38**, 1634–1658 (2021)

Comment 6: “Genome sequencing” and “Genome Size Estimation and Assembly” should be reorganized and clarified.

Response 6: We have carefully revised sections “Genome sequencing” and “Genome Size Estimation and Assembly” in the revised manuscript (Lines 634-679).

Comment 7: It is unclear to which species belong the Illumina DNAseq sequencing part (L608-611)

Response 7: Both of Illumina genomic libraries of *A. belladonna* and *D. stramonium* were constructed and sequenced on an Illumina HiSeq X Ten platform. We have revised this part in the revised manuscript (Lines 648-651).

Comment 8: RNA sequencing part for *L. chinense* is missing in Material and Method section

Response 8: Thanks for your kind suggestion. We have added the description for the RNA-seq of *L. chinense* in the Material and Method section of the revised manuscript (Lines 802-813).

Comment 9: BUSCO results for annotated genes are not shown

Response 9: We have performed the BUSCO analysis for annotated genes of *A. belladonna* and *D. stramonium* and provided those data in Supplementary Tables 7 and 8.

Comment 10: Data for *L. chinense* should also be made available on NCBI and figshare.

Response 10: We have uploaded the data of *L. chinense* on NCBI and figshare. The NCBI Bioproject and figshare link for raw data of *L. chinense* are PRJNA769498 and <https://figshare.com/s/9915396d9578595e0b05>, respectively (Lines 938 and 943).

Comment 11: The authors performed a very interesting and complete microsynteny analysis at the whole TA biosynthetic pathway scale to provide information of the evolution of the pathway. However, the whole organization of the pathway should be described in more details at the chromosome level. What about the respective position of each TA gene compared to other TA genes? Are there any true clusters besides the tropine cluster? Indicating TA genes on figure 1 or similar would be nice to see distribution.

Response 11: We have labeled the positions of each TA biosynthetic gene on the chromosome and provided those data in Supplementary Figs. 44 and 45, as well as the exact location of them in Supplementary Tables 24 and 25. We did not identify other true gene clusters associated with TA or mTA biosynthesis besides the tropine cluster we reported. Though two genes, *DsMPO.2* and *DsPMT.1*, are located close on chromosome LG10 of *D. stramonium*, the loci of *MPO* and *PMT* are not close in other

species of Solanaceae, such as *A. belladonna* and *P. axillaris*. Since the scope of this study is mainly on the evolution of TAs/mTAs biosynthesis, we did not present this *D. stramonium*-specific gene cluster in this study.

Supplementary Fig. 44. The positions of mTAs biosynthetic genes on the chromosomes of *A. belladonna*. The corresponding IDs and chromosomal positions of mTAs biosynthetic genes were provided in Supplementary Table 24.

Supplementary Fig. 45. The positions of mTAs biosynthetic genes on the chromosomes of *D. stramonium*. The corresponding IDs and chromosomal positions of mTAs biosynthetic genes in *D. stramonium* were provided in Supplementary Table 25.

Comment 12: What about the genes surrounding TA genes? Are there any interesting features? Can they represent putative TA genes?

Response 12: We retrieved 736 genes located 100 kb upstream and downstream of TAs biosynthetic genes from *A. belladonna* and *D. stramonium*. By surveying those genes, we did not find any interesting feature or novel gene that might be involved in TAs biosynthesis.

Comment 13: Based on their test of activity in planta, the authors indicated that DsLS displays a 16-fold higher activity compared to AbLS. Is this result supported by analysis of similar amount of both protein accumulation in the leaf extracts used for assays? (checked by Western blot for instance)?

Response 13: Thanks for your professional suggestions. The absence of an antibody against LS prevents us from checking LS protein accumulation in tobacco leaves by Western blot. Incidentally, attachment of the Flag tag at either the C- or the N-terminal of LS severely reduced and even abolished the LS activity of AbLS, as shown in the figure below, which is consistent with the discoveries in AsSCPL1 from oat and AtSMT from Arabidopsis that the enzymatic activity and accumulation of serine carboxypeptidase-like acyltransferase is very sensitive to the additional epitope tag (Mugford *et al.* 2009; Stehle *et al.* 2008). Consequently, we cannot monitor the accumulation of LS in tobacco leaves by detecting the attached tag using Western blot either.

To confirm the relative catalytic activity of AbLS and DsLS, we performed the experiments again. The relative AbLS and DsLS activities measured this time also support that DsLS activity is significantly higher than AbLS activity, approximately 17-fold. Admittedly, the ratio of DsLS activity to AbLS activity is similar to that in the original manuscript but they are not exactly the same. We thereby toned down the statement about DsLS activity by deleting “up to 16-fold” in the revised manuscript (Lines 443-444)

Reference:

- Mugford, S. T. *et al.* A serine carboxypeptidase-like acyltransferase is required for synthesis of antimicrobial compounds and disease resistance in oats. *Plant Cell* **21**, 2473–2484 (2009).
- Stehle, F., Stubbs, M. T., Strack, D. & Milkowski, C. Heterologous expression of a serine carboxypeptidase-like acyltransferase and characterization of the kinetic mechanism. *FEBS J.* **275**, 775–787 (2008).

The relative abundance of littorine in tobacco leaves infiltrated with *Agrobacterium* harboring DsLS-pEAQ, AbLS-pEAQ, AbLS-Flag-pEAQ (the flag tag was infused to the C-terminal of AbLS), and Flag-AbLS-pEAQ (the flag tag was infused to the N-terminal of AbLS).

Comment 14: The authors identified a P450 catalyzing N-demethylation of hyoscyamine. Albeit there is no doubt on the tests of activity they performed in both benth and yeast, one may imagine that this N-demethylase activity is not the primary of the enzyme (that could work with a higher efficiency on a distinct substrate). Have the authors considered the possibility to perform complementary analyses in planta by gene silencing for instance? albeit not mandatory, this would provide definitive evidences of the function of this enzyme.

Response 14: We overexpressed *EVM0022661* and knocked out it by CRISPR in *A. belladonna* hairy roots. We found that the norhyoscyamine level in *EVM0022661.2*-overexpressing hairy roots increased by ~2.09-fold to ~5.70-fold, compared with that in vector control lines. When *EVM0022661.2* was knocked out, the norhyoscyamine level decreased from ~17% to ~26% of that in control. Combined with the results generated from tobacco leave and yeast, our data indicated that *EVM0022661* is involved in the formation of norhyoscyamine. We have provided those data in the Fig. 6G and 6H of the revised manuscript and also attached it below. On the other hand, we cannot exclude the possibility that *EVM0022661.2* has additional enzyme activity with distinct substrates.

Figure 6G and 6H. The relative abundance of norhyoscyamine in EVM0022661.1 overexpressed lines (G) and edited lines (H). Error bars represent the SD of three biological replicates. **, $p < 0.001$; ***, $P < 0.0001$, Students' t-test.

Figures and tables:

Comment 15: Fig.3A species pictograms are not easy to read and should be replaced by species abbreviation to ease reading.

Response 15: We have added the species abbreviation in the Fig. 3A of the revised manuscript.

Comment 16: Please homogenize the figures S1, S2, S3

Response 16: We have provided homogenized Supplementary Figs. 1, 2, and 3 in the revised Supplementary Figures.

Comment 17: Showing all species on Fig. S4 would have been interesting. You could try UpSet plot from ComplexHeatmap package

Response 17: Thanks for your kind suggestion! We have performed the sharing of gene families using all species and presented this result by UpSet plot generated by the ComplexHeatmap package in the revised Supplementary Fig. S4.

Comment 18: X labels are missing on Fig. S6, Fig.S10B

Response 18: We have added those labels in the revised Supplementary Figs. 6 and 10.

Comment 19: Please provide a legend for Fig.S41, S42b, S43

Response 19: We have provided the legends in the revised Supplementary Figs. 42, 43, and 46 (the original Supplementary Figs. 41 42, and 43 were shifted to Supplementary Figs. 42, 43, and 46, respectively, because of the attachment of new supplementary figures).

Comment 20: Raw Illumina short-reads information is missing for *A. belladonna*. Similar information for all RNAseq data would be needed

Response 20: Thanks for your kind suggestion! We have uploaded raw illumine short-reads to NCBI SRA database and provided all the accession numbers of short-reads used in this study in Supplementary Table 26.

Other minor concerns:

Minor Comment 1: L49: angiosperm should be written Angiosperm

Response 1: We have used Angiosperm instead of angiosperm in the revised manuscript (Line 49).

Minor Comment 2: L60: eudicots \diamond Eudicots

Response 2: We have changed the eudicots to Eudicots in the revised manuscript (Line 61).

Minor Comment 3: L316-317 is unclear and should be rephrased.

Response 3: We have revised the sentence you mentioned in the revised manuscript (Lines 315-317). The revised sentence is “Taken together, the identified scopolamine biosynthetic genes and results of phylogenetic analysis combined with microsynteny analysis revealed the conservation of phenyllactylglucos and tropine biosynthetic genes in Solanaceae species.”.

Minor Comment 4: L629: “the full length transcriptomes were assembled” seems odd in a Genome assembly section

Response 4: The description of the full-length transcriptome assembly of *L. chinense* has been written as an independent paragraph in the revised manuscript (Lines 802-813).

Thank you for your helpful comments, which have helped us to improve our MS substantially.

Reviewer #2 (Remarks to the Author):

The authors report the chromosome scale genome assemblies of the tropane alkaloid producing species *Atropa belladonna* and *Datura stramonium*. The authors utilize these assemblies to investigate the evolution of tropane alkaloid biosynthesis within the Solanaceae family. Not all members of the Solanaceae synthesize the medicinal tropane alkaloids reported in this manuscript, which creates the potential to investigate the evolutionary trajectory of these metabolites. The topic is of broad interest to researchers interested in plant specialized metabolism. The genomic resources will also be useful for biologists interested in comparative biology of the Solanaceae family. However, there are a number of concerns throughout the manuscript relating to data quality, interpretation, presentation, and availability that need to be addressed prior to the manuscript being suitable for publication.

Response:

Thank you very much for these supportive comments on our work. We really appreciate all your valuable suggestions on our MS. We present point-to-point responses below.

Major Concerns

Comment 1: As presented, the WGCNA data presented in Figure S40 and S41 is uninformative. At a minimum, a supplemental dataset file (Excel sheet) should be presented that provides gene identifier information and annotation for the genes in each module. In the current form, the data are not useful for the reader.

Response: We have provided an excel sheet containing the gene IDs and their annotation in each module in Supplementary Data 1.

Comment 2: The resolution of many of the supplemental figures is extremely poor, especially the synteny figures (S26 through S38 and S42). The text is too small and does not allow for the reading of the gene identifiers. When the reader zooms in the images are blurred. In addition, in the case of all of the synteny figures, reference should be made to the corresponding chromosome numbers that are referenced for each species so that the reader can clearly follow the syntenic regions being discussed.

Response 2: We have improved the figure quality in the revised version. By the way, according to the kind suggestions given by reviewers, Supplementary Figures 1, 2, and

3 are homogenized. The corresponding chromosome numbers that are referenced for each species have been added to all of the synteny figures (Supplementary Figs. 26-34, 36-38, and 51).

Comment 3: There is a general lack of detail and information provided for most of the supplemental figures and supplemental tables. In many cases, legends are inadequate and do not provide the reader with necessary information to interpret the figure or the data.

Response 3: Thanks for your kind suggestion. We have provided detailed information in the revised supplementary figures and tables.

Comment 4: Figure S1 is confusing. The shape of the curves for the two species is very different and the y-axes of each graph differs. Why is there this difference. Information should be provided that allows the reader to understand how the genome size of each species can be inferred from this k-mer analysis.

Response 4: We have redrawn this figure in the revised Supplementary Fig. 1. As per your suggestion, we provided the detailed methods of *K*-mer analysis as well in the legends of the revised Supplementary Fig. 1.

Comment 5: Figure S2b: In the contact map for *D. stramonium*, there appears to be small regions between LG4 & LG5 and LG9 & LG10 that are not necessarily part of either linkage group and show signal with other regions of the genome. Can the authors provide an explanation for these data?

Response 5: We checked the Hi-C assembly again and found that the region you mentioned is in the range of 128900000 bp to 129100000 bp in LG04 and 80800000 bp to 81000000 bp in LG09. We checked the assembly of scaffolds from contigs and found that these potential miss-linked regions respectively located in one single contig. In addition, we used Pacbio long reads to map the assembly genome to validate the Hi-C contact map and found that the regions are covered with Pacbio long reads, as shown below. It is thereby unlikely that the two regions are miss-linked.

Alignment Pacbio long reads to assemble the genome of *D. stramonium*. The figure showed the 128900000 bp to 129100000 bp of LG04.

Alignment Pacbio long reads to assembly genome of *D. stramonium*. The figure showed the range of 80800000 bp to 81000000 bp in LG09

Comment 6. Figure S3: There appears to be two GC peaks for the *D. stramonium* data. Why is this? Could it be associated with the transposable element content? Some of the graph axes do not have labels.

Response 6: We carefully analyzed GC contents in the *D. stramonium* genome again to interpret the bimodal GC distribution. As shown in the figures below, we retrieved the corresponding regions around the two GC peaks and found that the low-GC content region has lower transposable element content whereas the high-GC content region has higher transposable element content. Thus, the bimodal GC distribution is, as you supposed, associated with the transposable element content, especially the Gypsy of

LTR, which expanded recently in *D. stramonium*. In addition, we have revised Supplementary Fig. 3 as per the other reviewers' suggestions to unify the styles of the figures.

The diagram of the high-GC region and low-GC region in the *D. stramonium* genome. The high-GC region was highlighted with a green rectangle, while the low-GC region was highlighted with a red rectangle.

The composition of TEs in high-GC fragments (left) and low-GC fragments (right) of the *D. stramonium* genome.

Comment 7: Figure 4 and associated manuscript text regarding the evolution of tropinone reductases: The authors should consider in their interpretation of their data that the tomato and potato TRI enzymes may actually have undergone neofunctionalization and that tropinone may not be their preferred substrate in planta. These genes have been retained in these species and may now have distinct biochemical activities that are yet undefined.

Response 7: We agree with you that it is hard to reject the speculation that the *TRI* from tomato and potato might have undergone neofunctionalization and may have distinct biochemical activities by a gain of a novel enzymatic function. We have discussed this in the revised manuscript (Lines 586-590).

Comment 8: Figure S43: The authors should provide information regarding the sources of the gene expression data they are using for tomato, potato, and pepper.

Response 8: Thanks for your kind suggestion! We have added the citation information regarding the sources of the gene expression data used in the revised Supplementary Fig. 46. Supplementary Fig. 43 has been shifted to 46 because of the attachment of new supplementary figures. The raw RNA-Seq data used for analyzing the expression of *CYP82M3* and *TRI* in *C. annuum* (Qin et al. 2014), *S. tuberosum* (Xu et al. 2011), and *S. lycopersicum* (Kumar et al. 2021) was downloaded from NCBI SRA database with accession number SRP019256 for *C. annuum*, SRA030516 for *S. tuberosum*, and SRP229637 for *S. lycopersicum*.

Reference:

- Qin, C. et al. Whole-genome sequencing of cultivated and wild peppers provides insights into Capsicum domestication and specialization. *Proc. Natl. Acad. Sci. U. S. A.* **111**, 5135–5140 (2014).
- Xu, X. et al. Genome sequence and analysis of the tuber crop potato. *Nature* **475**, 189–195 (2011).
- Kumar, V. et al. Identification of tomato root growth regulatory genes and transcription factors through comparative transcriptomic profiling of different tissues. *Physiol. Mol. Biol. Plants* **27**, 1173–1189 (2021).

Comment 9: The *D. stramonium* contig number provided in Table 1 does not match the number provided in Table S2.

Response 9: Thank you for catching these mistakes, which have been corrected in Table 1 of the revised manuscript and Supplementary Table 2.

Comment 10: The assembled genome size for *A. belladonna* provided in Table 1 does not match the figure provided in Table S2 and neither of these numbers match the assembled genome size provided on line 146 of the manuscript file. The authors should check these data.

Response 10: Thank you for catching these mistakes. We have carefully checked and corrected the data in line 146, Table 1 of the revised manuscript, and Supplementary Table 2.

Comment 11: In Table S4 the bulk of the *D. stramonium* genome assembly appears to be located in 85 contigs. However, depending on whether the contig number in Table 1 or Table S2 is correct, it means there are approximately 1000 contigs totaling close to 50 Mb that are not in the assembly. Can the authors provide some insight into this discrepancy.

Response 11: According to your kind comments 9 and 10, we have revised those data as we described in responses 9 and 10. Admittedly, 85 contigs composed, as you mentioned, the bulk of the *D. stramonium* genome. In other words, 85 out of 1298 contigs were anchored to the chromosomes. To interpret this discrepancy, we compared the length of contigs anchored to chromosomes with that of contigs that were not anchored to the chromosomes. As shown below, the contigs anchored to chromosomes have a significantly larger size than those were not anchored to the chromosomes. The median of the former is 2,580,420 bp, while the latter is 34,069 bp. Thus, the longer contigs are more likely anchored to chromosomes. In addition, this result is consistent with the results published by Sharma et al. They found that the development of highly accurate long reads by repeated sequencing of circularized DNA (HiFi; PacBio) has greatly increased the size of contigs, only 30 larger contigs of 779 total contigs provided a good assembly for the *Macadamia janseni* genome (Sharma et al. 2022).

The comparison between the length of contigs anchored to chromosomes and that of contigs not anchored to chromosomes. The y-axis represents the length of the contigs. The plot represents the median with 95% CI.

Reference:

Sharma, P., Masouleh, A. K., Topp, B., Furtado, A. & Henry, R. J. *De novo* chromosome level assembly of a plant genome from long read sequence data. *Plant J.* **109**, 727–736 (2022).

Comment 12: Table S5 and S6: No footnotes or explanations are provided for these tables so the data is difficult to interpret. However, it is extremely unusual in the case of the *de novo* and the homology based annotations that many of the parameters reported have identical values in each table for the two different species. For example, it seems highly unlikely that the average intron length predicted by AUGUSTUS for both species be exactly the same value. The authors are asked to thoroughly check the data in these tables. Furthermore, the average gene length as predicted by Genscan seems particularly high for both species.

Response 12: We apologize for our carelessness when we prepared those sheets and we have corrected those data in the revised Supplementary Tables 5 and 6. For the issue of the average length of gene predicted by Genscan, we first checked the raw annotation file and no calculation error was found. Subsequently, we found that this situation not occurred solely in this study. In the eggplant genome (Wei et al. 2020) and carrot genome (Iorizzo et al. 2016), the average transcript length predicted by Genscan is the longest among that predicted by other software. The average transcript length predicted by Augustus is 3,527.60 bp in the eggplant genome, while that predicted by Genscan is 16,795.18 bp (Wei et al. 2020, Supplementary Table 3). Similarly, the average transcript length predicted by Augustus and Genscan is 2,781 bp and 8,015 bp, respectively (Iorizzo et al. 2016, Supplementary Table 16). Therefore, we think this is due to the characteristics of Genscan.

Reference:

Wei, Q. *et al.* A high-quality chromosome-level genome assembly reveals genetics for important traits in eggplant. *Hortic. Res.* **7**, (2020).

Iorizzo, M. *et al.* A high-quality carrot genome assembly provides new insights into carotenoid accumulation and asterid genome evolution. *Nat. Genet.* **48**, 657–666 (2016).

Comment 13: The data provided in Figure 3C may not be accurate as it relates to the evolution of the PPAR enzyme. For example, the authors have performed end-point assays containing very large amounts of recombinant protein (20 micrograms) and 2 millimolar phenylpyruvic acid. PPAR belongs to a family of enzymes that are

catalytically promiscuous and it is possible that the authors are simply forcing a biochemical reaction to occur by utilizing large amounts of enzyme and substrate. The authors should be careful about drawing evolutionary conclusions from such experiments and would be better advised to perform kinetic analyses of these recombinant enzymes as they have completed for the TRI enzymes from different Solanaceae species. In addition, the specific gene IDs should be provided for each PPAR enzyme that was characterized.

Response 13: We have performed enzymatic kinetic analyses of these PPARs from Solanaceae as per your kind suggestion. We provided the Michaelis–Menten curves in Supplementary Fig. 39 and enzymatic parameters in Supplementary Table 29. We also provided the gene IDs as per your recommendation in the revised manuscript (Lines 356-358), as well as the primers we used in Supplementary Table 26.

Supplementary Fig. 39. Michaelis–Menten curves for the AbPPAR (a), DsPPAR (b), CaPPAR (c), SIPPAR (d), and PaPPAR (e). The buffer for the reduction reaction was potassium phosphate (50 mM, pH 8.0). Phenylpyruvic acid was used as substrate. Each point is the mean of triplicate assays.

Comment 14: The data describing the characterization of the hyoscyamine N-demethylase would be strengthened if RNAi or VIGS data was provided to support the enzyme activity data.

Response 14: We established the hyoscyamine *N*-demethylase, *EVM0022661.2*, overexpressed and edited transgenic *A. belladonna* hairy roots. We found that the norhyoscyamine level in *EVM0022661.2* overexpressed hairy root increased by ~2.09-fold to ~5.70-fold, compared with that in vector control lines. When *EVM0022661.2* was edited, the norhyoscyamine level decreased from ~17% to ~26% of that in control. Combined with the results generated from tobacco leave and yeast, our data indicated that *EVM0022661* indeed participated in the norhyoscyamine. We provided the relative abundance of norhyoscyamine in *EVM0022661.2* overexpressed lines and edited lines in Fig. 6G and 6H. We also provided the results of genomic PCR and qRT-PCR in transgenic *A. belladonna* hairy root in Supplementary Fig. 49 and the primers used in this experiment in Supplementary Table 30.

Fig. 6G and 6H. The relative abundance of norhyoscyamine in *EVM0022661.1* overexpressed lines (G) and edited lines (H). Error bars represent the SD of three biological replicates. **, $p < 0.001$; ***, $P < 0.0001$, Student's *t*-test.

Comment 15: No table is provided that reports all of the primers that were utilized for gene cloning and construct assembly.

Response 15: Thanks for your kind suggestion. We have provided all the primers used in this study as Supplementary Tables 25-30.

Comment 16: The manuscript would benefit from English language editing.

Response 16: The language of the revised manuscript has been polished by Springer Nature Author Services.

Minor comments:

Minor Comment 1: Figure S10 b through e: The units on the x-axes for the graphs are not defined.

Response 1: We have added the units on the x-axes, Mya, million years ago, in the revised Supplementary Fig. 10.

Minor Comment 2: Figure S19: The gene name is tropinone reductase I. Not phenyllactate tropine forming reductase.

Response 2: Thanks a lot for catching this mistake, which has been corrected in the revised Supplementary Fig. 19.

Minor Comment 3: Figure S39: Gene name should be UGT1.

Response 3: We have corrected this in the revised Supplementary Fig. 40 (The original Supplementary Fig. 39 has shifted to Supplementary Fig. 40 because of the addition of new supplementary figures).

Minor Comment 4: Table S10: Note footnotes provided. The abbreviation TRF is not defined.

Response 4: We have defined the TRF in the footnotes of the revised Supplementary Table 12 (The original Supplementary Table 10 has been shifted to 12 because of the addition of new supplementary tables).

Minor Comment 5: Manuscript text lines 67 -68: The statement that not all tropanes have physiological activity is not entirely accurate. It's correct that some tropanes have very strong pharmaceutical activities. However, there are many tropanes where their bioactivities are simply not studied and the physiological roles are therefore not defined. The text should be edited accordingly.

Response 5: Thanks for your kind suggestion. We reworded this sentence in the revised manuscript as “However, not all TAs have defined physiological activity.” (Lines 67-68).

Minor Comment 6: Lines 82 and 83: UGT1 does not catalyze the formation of phenyllactate UDP glucose. It catalyzes the formation of phenyllactoyl glucose. UDP is a co-product of the reaction.

Response 6: We have used phenyllactylglucose instead of phenyllactoyl UDP-glucose in the revised manuscript (Lines 82-83).

Minor Comment 7: Line 270: please define HPD.

Response 7: We have defined HPD (highest posterior density) in the revised manuscript (Line 271).

Minor Comment 8: Figure 3D should read PPAR not PPR.

Response 8: We have corrected this mistyping in the revised Fig. 3.

Minor Comment 9: Line 768: I believe the vector name is pHannibal

Response 9: Thanks for your professional comments. We have corrected this mistyping in the revised manuscript (Line 823).

Special thanks to you for your good comments sincerely.

Reviewer #3 (Remarks to the Author):

The manuscript entitled „The genomes of *Atropa belladonna* and *Datura stramonium* reveal the evolution of tropane alkaloid biosynthesis in the Solanaceae “ by Zhang and colleagues uses the newly assembled genomes of *Atropa belladonna* and *Datura stramonium* to shed light on the tropane alkaloid biosynthesis.

In total the authors have produced a solid manuscript using state of the art methods.

The main interesting findings were

- 1) chromosome level assemblies of *A. belladonna* and *D. stramonium*
- 2) Using hairy root transformation to show that an *A. belladonna* root isoform of MPO is likely the active isoform in roots
- 3) Showing that the duplication of TRI potentially allowed for more flexibility as this allowed a more root specific isoform as well as an enzyme better adapted to TRI activity. Surprisingly, also the authors could show that tomato TRI is active but at a very low level.
- 4) Revealing information about LS genes

Novelty

This report shows interesting information about the phylogeny and potentially evolution of tropane biosynthesis.

Response:

Thank you very much for your positive comments. We really appreciate all your valuable suggestions on our MS. We present point-to-point responses below.

Comments

Comment 1: Since there are two isoforms of AbMPO and one is root expressed wouldn't a root specific (i.e. hairy culture) downregulation of the less root expressed isoform abMPO2 be expected to have no major outcome? I would thus tone down the statement about MPO responsibility and/or also indicate that tropanes are usually synthesized in the roots.

Response 1: Our results, as you mentioned, cannot exclude the possibility that AbMPO2 may participate in the biosynthesis of TAs. One of the reasons that promote us to draw this conclusion is that TAs are usually synthesized in the roots, yet we did not declare this in the original manuscript. We have added information about the tissues where mTAs were biosynthesized in the revised manuscript (Lines 87-90). And toned down the statement about AbMPO1 by saying “Thus, AbMPO1 is the primary functional MPO involved in mTA biosynthesis in *A. belladonna*.” in the revised manuscript (Line 294).

Comment 2: Based on Figure 4 would the presence of multiple TRI genes in Capsicum and Datura not potentially be also explicable by a duplication of TRI that occurred in the Solanoideae [Särkinen 2013: comprising major Atropina, Solaneae, Capsiceae and Physaleae + small like Datureae] and lost in Solaneae (or the Solanum genus)? One could check easily in the Physalis genome even though a larger sampling would be interesting anyway.

Response 2: Thanks for your suggestion! We have checked the *TRI* duplication in *Physalis floridana* as per your recommendation. There are two tandem duplicated *TRIs* in the *Physalis* genome published by Lu et al. In addition, the *CYP82M3* homolog in *Physalis* is localized beside *TRI* as well. Those results support our conclusion that this gene cluster is widespread distribution in Solanaceae (as shown below). Furthermore, Kubwabo et al. reported that *Physalis peruviana*, another species in the *Physalis* genus, contains tropine. Thus, we agree with your opinion that the duplication of *TRI* might occur in the Solanoideae and be lost in some genera in Solaneae. We have added the microsynteny analysis results in Supplementary Fig. 51 and discussed this hypothesis in the discussion section of the revised manuscript (Lines 580-586).

Supplementary Fig. 51. Microsynteny analysis of TRI and CYP82M3 genes between *A. belladonna* (Abe) and *P. floridana* (Pfl). The syntenic *CYP82M3* genes were highlighted with red and the syntenic *TRI* genes were highlighted with green. The genomic region and chromosome ID were placed under the abbreviation of species names.

Reference:

- Lu, J. *et al.* The *Physalis floridana* genome provides insights into the biochemical and morphological evolution of *Physalis* fruits. *Hortic. Res.* **8**, (2021).
- Kubwabo, C., Rollmann, B. & Tilquin, B. Analysis of alkaloids from *Physalis peruviana* by capillary GC, capillary GC-MS, and GC-FTIR. *Planta Med.* **59**, 161–163 (1993).

Comment 3: It is good that the raw reads are publicly available in NCBI. However, the assembly with at least gene annotations must be made available in an INSDC database or a China CNCB resource and not on figshare to allow follow up studies

Response 3: Thanks for your kind suggestion. We have updated the assembly and annotations of *A. belladonna* and *D. stramonium* to CNCB under accession numbers GWHBOWM000000000 and GWHBOZL000000000, respectively and provide that information in the revised manuscript (Lines 943-946).

Comment 4: It would have been nice to speculate on the role of tomato TRI and to investigate which potential regions in the enzyme could lead to the difference in activity.

Response 4: We have compared the sequences of TRI from TAs-producing and non-TAs-producing species (Supplementary Fig. 48), and found that Leu 159 might contribute to the low or non-TRI activity in SITRI and StTRI. Subsequently, we analyzed the TRI activity of SITRI^{L159V} and StTRI^{L159V}. L159V substitution in SITRI dramatically increased the enzymatic activity of catalyzing tropinone to tropine. More interesting is that the same amino acid substitution in TRI from potato makes it gain the catalytic capacity to transfer tropinone to tropine (Fig. 4). We have provided those data in Fig. 4C, 4E, and 4F of the revised manuscript (Lines 399-407).

Comment 5: when describing genome sizes say e.g. tomato and tobacco as there are maybe larger genomes in the *Cyphomandra* genus than tobacco - just rewording is enough

Response 5: Thanks for your professional suggestion. As you recommended, we searched the genome sizes in *Cyphomandra*. None of the species in this genus have been sequenced. However, according to earlier research on this genus by Pringle GJ, nuclear DNA amounts in *Cyphomandra* are the largest in the Solanaceae family. The range of 2C nuclear DNA amounts of *Cyphomandra* spp is 13.5 - 49.6 pg, which were measured by flow cytometry, while the owner of the second largest 2C nuclear DNA

amounts is *Nicotiana* spp., about 6.04 - 6.91 pg. We have reworded this sentence in the revised manuscript (Line 225).

Reference:

Pringle, Gregory James. The cytogenetics of the tamarillo, *Cyphomandra betacea* (Cav.) Sendt., and its wild relatives. The University of Auckland (New Zealand), 1991.

Comment 6: line 225 gypsy expansion give citation or qualify statement

Response 6: We have cited five researches in which the authors found that the gypsy expansion contributed to the genome expansion of species in Solanaceae in the revised manuscript (Line 227), including:

32. Kim, S. S. B. S. *et al.* New reference genome sequences of hot pepper reveal the massive evolution of plant disease-resistance genes by retroduplication. *Genome Biol.* **18**, 1–11 (2017).
33. Qin, C. *et al.* Whole-genome sequencing of cultivated and wild peppers provides insights into *Capsicum* domestication and specialization. *Proc. Natl. Acad. Sci. U. S. A.* **111**, 5135–5140 (2014).
34. Lu, J. *et al.* The *Physalis floridana* genome provides insights into the biochemical and morphological evolution of *Physalis* fruits. *Hortic. Res.* **8**, (2021).
35. Bolger, A. *et al.* The genome of the stress-tolerant wild tomato species *Solanum pennellii*. *Nat. Genet.* **46**, 1034–1038 (2014).
36. Wu, M., Kostyun, J. L. & Moyle, L. C. Genome sequence of *Jaltomata* addresses rapid reproductive trait evolution and enhances comparative genomics in the hyper-diverse Solanaceae. *Genome Biol. Evol.* **11**, 335–349 (2019).

Comment 7: line 244 I wouldn't call the *Atropa* WGT species specific as close relatives were not investigated--- just wording

Response 7: We have deleted the “species-specific” in that sentence to make this statement more precise and accurate in the revised manuscript (Line 245).

We appreciate your helpful comments, thank you very much.

Reviewers' Comments:

Reviewer #1:

Remarks to the Author:

In the revised version of their manuscript, Zhang and coworkers addressed almost all the concerns raised by my previous reviewing. They notably provided compelling evidences of the in planta role of their P450 by performing overexpression and knock out of the corresponding gene. Only a few concerns remain as listed below:

- Was pilon used only once for *D. stramonium* (when used 4 times for *A. belladonna*)?
- We have no idea of the quality of *L. chinense* transcriptome assembly. Less genes have been found in *L. chinense* compared to both genomes. Does this impact the BUSCO score for instance?
- RNAseq analysis is not consistent:
- You used SeqClean and PASA for transcriptome assembly (LL719-722) but TRIMMOMATIC and TRINITY (LL787-789).
- "Fragments per kilobase of exon model per million reads mapped (FPKM) values for RNA-Seq reads were calculated with HISAT2 (v.2.0.5) and CUFLINKS (v 2.2.1) with default parameters. The read counts extracted from StringTie..." Was StringTie of HISAT/Cufflinks used? Please note that cufflinks is considered depreciated.
- Also, it is unclear whether you used de novo assembled transcripts or Genome-annotated genes for your differential analysis.
- R package versions are still missing in the revised manuscript (including the R version used).

Reviewer #2:

Remarks to the Author:

The authors have addressed the comments made by the reviewers on the original submission of this manuscript. I only have minor comments left to address:

- 1) Please check with the editor. The use of red and green for distinguishing the genes in Figure 4A may be problematic for color blind readers.
- 2) Lines 57 & 58: Please check for correctness. I do not believe that these metabolites are nerve agent poisons. I think they can be used as antidotes to nerve agents.
- 3) Line 94: change to intermediate
- 4) Line 144: provide the fold coverage in parentheses associated with the 52.47 Gb
- 5) Line 263: please check for accuracy. Should this be "inactive" rather than "active"
- 6) Line 317: change to phenyllactoylglucose.
- 7) Lines 533 and 534: The gene number in *A. belladonna* is larger than in *D. stramonium* due to polyploidy
- 8) Line 535: I don't think it's unexpected that an increase in TE number results in increased genome size. I think this is well known.
- 9) Line 536 & 537: This sentence doesn't make sense. I think what the authors are trying to say is that the scopolamine related genes have not all been characterized in a single species.
- 10) Line 549: I think the authors mean to detoxify metabolites, not cells.
- 11) Line 562 & 563: I don't necessarily agree with this statement. The data actually suggest that the biosynthesis of tropine is not conserved across the family. For example, Figure 4B, tropine is not detected in petunia, potato, tomato or tobacco. Please revise this sentence.
- 12) Line 564: change spice to capsaicin
- 13) Line 630: should be *P. hybrida*.
- 14) Line 632: italicize *Nicotiana benthamiana*
- 15) Line 654 – 655: use italics appropriately for restriction enzymes.
- 16) Line 834: there is no mention of an internal standard being incorporated into the extraction solvent. Please provide this information. If no internal standard was utilized, it will seriously compromise the ability of the authors to accurately normalize and quantify their mass-spec data.
- 17) Line 858: change to described.

- 18) Line 870: respectively
- 19) Line 877: change to described.
- 20) Line 914: Italicize the "N"
- 21) Check the reference list for accuracy. The author is listed incorrectly for reference 11 and reference 12 is duplicated again in reference 111. There may be other errors I didn't catch.
- 22) Supplemental Fig 1 legend line 5: "the" is duplicated
- 23) Supplemental Figure 11 – Change title to "the characterization of AbMPOs"
- 24) Supplemental Figures 44 and 45: change PPR to PPAR
- 25) Supplemental Figure 48: The legend refers to a dashed black box but the figure shows a red box. Please update the legend.

Reviewer #3:

Remarks to the Author:

The authors have adequately and exhaustively addressed the comments raised. A big thank you for also making all data available.

Reviewer #1:

In the revised version of their manuscript, Zhang and coworkers addressed almost all the concerns raised by my previous reviewing. They notably provided compelling evidences of the in planta role of their P450 by performing overexpression and knock out of the corresponding gene. Only a few concerns remain as listed below:

Response:

Thank you very much for these positive comments on our work. We really appreciate all your valuable suggestions which improved this MS significantly. We present point-to-point responses below.

Comment 1: Was pilon used only once for *D. stramonium* (when used 4 times for *A. belladonna*)?

Response 1: The distinct assembly strategy used in this study is due to the different sequencing methods employed. We employed Nanopore technology to sequence the *A. belladonna* genome while HiFi to sequence *D. stramonium*. Nanopore sequencing has higher error rates (at 6–15%) than HiFi sequencing (< 1%) (Chen *et al.*, 2021). Thus, most genome assembly based on Nanopore ONT sequencing data employed more rounds of polishing (Cheng *et al.*, 2021; Yang *et al.*, 2021) compared with that based on HiFi reads (Zhang *et al.*, 2020; Ma *et al.*, 2021). In addition, although we used two distinct strategies to polish, the short reads mapping rates and coverage of the *A. belladonna* and *D. stramonium* genome are both over 99% (Supplementary Table 2), indicating our assembly strategies are valid.

Reference:

Chen Y, Zhang Y, Wang AY, Gao M, Chong Z. 2021. Accurate long-read de novo assembly evaluation with Inspector. *Genome Biology* **22**: 312.

Cheng J, Wang X, Liu X, Zhu X, Li Z, Chu H, Wang Q, Lou QQ, Cai B, Yang Y, et al. 2021. Chromosome-level genome of Himalayan yew provides insights into the origin and evolution of the paclitaxel biosynthetic pathway. *Molecular Plant* **14**: 1199–1209.

Ma D, Dong S, Zhang S, Wei X, Xie Q, Ding Q, Xia R, Zhang X. 2021. Chromosome-level

reference genome assembly provides insights into aroma biosynthesis in passion fruit (*Passiflora edulis*). *Molecular Ecology Resources* **21**: 955–968.

Yang X, Gao S, Guo L, Wang B, Jia Y, Zhou J, Che Y, Jia P, Lin J, Xu T, et al. 2021. Three chromosome-scale *Papaver* genomes reveal punctuated patchwork evolution of the morphinan and noscapine biosynthesis pathway. *Nature Communications* **12**: 6030.

Zhang L, Chen F, Zhang X, Li Z, Zhao Y, Lohaus R, Chang X, Dong W, Ho SYW, Liu X, et al. 2020. The water lily genome and the early evolution of flowering plants. *Nature* **577**: 79–84.

Comment 2: We have no idea of the quality of *L. chinense* transcriptome assembly. Less genes have been found in *L. chinense* compared to both genomes. Does this impact the BUSCO score for instance?

Response 2: The BUSCO analysis of the transcriptome of *L. chinense* generated 72% of complete BUSCOs. Yet it is lower than the complete BUSCOs based on the genomes of *A. belladonna* and *D. stramonium*, we just used it to infer the phylogenetics of *A. belladonna* and *D. stramonium* in the Solanaceae family. Thus, we think the completion of *L. chinense* transcriptome is qualified for this work. The bootstrap values of the phylogenetic tree in Fig. 2 are all 100%. To further validate the robustness of the inferred species tree, we used genome annotated genes of *Lycium barbarum*, a species in the same genus as *L. chinense*, to build the phylogenetic tree and got the phylogenetic tree with the same topology (Fig. 1). In addition, we have added the BUSCO information of *L. chinense* transcriptome in the revised supplemental table 17.

Fig. 1 The ML trees were constructed with 1000 bootstrap values using RAxML. A, the ML tree built based on the genome annotated genes of *S. lycopersicum*, *S. tuberosum*, *S. melongena*, *C. annuum*, *D. stramonium*, *L. barbarum*, *A. belladonna*, *N. tabacum*, *P. axillaris*, *C. canephora*, *M.*

guttatus, *V. vinifera* and *O. sativa*. B, The ML tree was built based on the genome annotated genes of species used in A and the transcriptome of *L. chinense*. Bootstrap support rates (%) for the branches are indicated by the numbers next to the branches. *A. belladonna*, *L. barbarum*, and *L. chinense* were highlighted with light red, blue and green, respectively.

Comments 3: RNAseq analysis is not consistent:

- You used SeqClean and PASA for transcriptome assembly (LL719-722) but TRIMMOMATIC and TRINITY (LL787-789).
- " Fragments per kilobase of exon model per million reads mapped (FPKM) values for RNA-Seq reads were calculated with HISAT2 (v.2.0.5) and CUFFLINKS (v 2.2.1) with default parameters. The read counts extracted from StringTie..." Was StringTie of HISAT/Cufflinks used? Please note that cufflinks is considered depreciated.
- Also, it is unclear whether you used de novo assembled transcripts or Genome-annotated genes for your differential analysis.
- R package versions are still missing in the revised manuscript (including the R version used).

Response 3: Thanks to your careful observations, we have revised them accordingly. We used genome-annotated genes, instead of de novo assembled transcripts, for differential analysis and therefore we did not use TRINITY to de novo assemble the transcript based on RNA-seq data. To your concern about Cufflinks, we used StringTie but not Cufflinks to extract the read counts. In addition, transcripts per million (TPM) instead of FPKM was used to analyze the relative expression of genes and WGCNA analysis. We have rewritten this part of the manuscript. Additionally, we updated the revised manuscript to include R versions and packages, as your kind suggestion. (Lines 790-799)

Thank you for your helpful comments, which have helped us to improve our MS substantially.

Reviewer #2 (Remarks to the Author):

The authors have addressed the comments made by the reviewers on the original submission of this manuscript. I only have minor comments left to address:

Response:

Thank you very much for these positive comments on our work. We appreciate all your valuable suggestions on our MS. We present point-to-point responses below.

Comment 1: Please check with the editor. The use of red and green for distinguishing the genes in Figure 4A may be problematic for color blind readers.

Response 1: Thank you for your kind advice. We have changed the two colors in Figure 4A into blindness-safe colors in the revised manuscript.

Comment 2: Lines 57 & 58: Please check for correctness. I do not believe that these metabolites are nerve agent poisons. I think they can be used as antidotes to nerve agents.

Response 2: We have corrected this issue by substitution of “poisons” for “antidotes” in the revised manuscript (Lines 57 & 58).

Comment 3: Line 94: change to intermediate

Response 3: We have changed “intermedia” to “intermediate” in the revised manuscript (Line 94).

Comment 4: Line 144: provide the fold coverage in parentheses associated with the 52.47 Gb

Response 4: We have added the fold coverage of HiFi reads in the revised manuscript (Line 144).

Comment 5: Line 263: please check for accuracy. Should this be “inactive” rather than “active”

Response 5: In the revised manuscript, we have replaced the sentence with a more accurate and detailed statement to eliminate the distortion of our results. Thanks again for your thoughtful comments (Lines 263-266).

Comment 6: Line 317: change to phenyllactoylglucose.

Response 6: We have corrected this mistake in the revised manuscript by changing it to phenyllactylglucose. It is a little different from your suggestion, “phenyllactoylglucose” because though either of them can represent glycosylated phenyllactate, we employed phenyllactylglucose in this manuscript (Line 319).

Comment 7: Lines 533 and 534: The gene number in *A. belladonna* is larger than in *D. stramonium* due to polyploidy

Response 7: We have revised the statement you mentioned in the revised manuscript (Line 536).

Comment 8: Line 535: I don’t think it’s unexpected that an increase in TE number results in increased genome size. I think this is well known.

Response 8: We completely agree with you that TE burst is the common cause of increased genome size. We have deleted the word “unexpected” in the revised manuscript (Line 537).

Comment 9: Line 536 & 537: This sentence doesn’t make sense. I think what the authors are trying to say is that the scopolamine related genes have not all been characterized in a single species.

Response 9: We have corrected this sentence in the revised manuscript. Consistent with your speculation, what we are trying to convey is that these genes are not all

characterized in a single species. (Line 539).

Comment 10: Line 549: I think the authors mean to detoxify metabolites, not cells.

Response 10: We have substituted “cells” with “metabolites” in the revised manuscript (Line 551).

Comment 11: Line 562 & 563: I don’t necessarily agree with this statement. The data actually suggest that the biosynthesis of tropine is not conserved across the family. For example, Figure 4B, tropine is not detected in petunia, potato, tomato or tobacco. Please revise this sentence.

Response 11: We completely agree with you that the statement may lead readers to a point that is contrary to the data we presented and what we really think. We have changed this sentence to “...that the biosynthetic genes of tropine are conserved across this family.” because though we cannot detect tropine in petunia, potato, tomato, or tobacco, the biosynthetic genes (active or inactive) can be found in their genome (Line 564 & 565).

Comment 12: Line 564: change spice to capsaicin

Response 12: We have changed “spice” to “capsaicin” in the revised manuscript (Line 566).

Comment 13: Line 630: should be *P. hybrida*.

Response 13: We have changed “*P. hybrid*” to “*P. hybrida*” in the revised manuscript (Line 632).

Comment 14: Line 632: italicize *Nicotiana benthamiana*

Response 14: We have italicized *Nicotiana benthamiana* in the revised manuscript (Line 634)

Comment 15: Line 654 – 655: use italics appropriately for restriction enzymes.

Response 15: We have italicized those restriction enzymes in the revised manuscript (Lines 656 & 657).

Comment 16: Line 834: there is no mention of an internal standard being incorporated into the extraction solvent. Please provide this information. If no internal standard was utilized, it will seriously compromise the ability of the authors to accurately normalize and quantify their mass-spec data.

Response 16: To quantify scopolamine and its intermediates, some groups have a great tradition to use internal standards (*The Plant Cell* 2014; *Nature Communications* 2018). And recently, many groups have established new methods by using external standards (*Nature Communications* 2018; *ACS Synthetic Biology* 2019a, b; *Nature* 2020; *PNAS* 2021). Both methods now have been recognized by most experts in this field. By using external standards, we have established a precise and stable method (*Organic Letters* 2018; *New Phytologist* 2020; *ACS Catalysis* 2021). We have extracted our quantification data to build the standard curve for main compounds (Fig. 2). As you can see, our method is adequate.

Figure 2, The standard curves of hyoscyamine, Littorine, scopolamine, phenyllactic acid and norhyoscyamine measured by LC-MS.

References:

Bedewitz MA, Góngora-Castillo E, Uebler JB, Gonzales-Vigil E, Wiegert-Rininger KE, Childs KL, Hamilton JP, Vaillancourt B, Yeo Y-S, Chappell J, et al. 2014. A Root-Expressed L-Phenylalanine: 4-Hydroxyphenylpyruvate Aminotransferase Is Required for Tropane Alkaloid Biosynthesis in *Atropa belladonna*. *The Plant Cell* 26: 3745–3762.

Bedewitz MA, Jones AD, D'Auria JC, Barry CS. 2018. Tropinone synthesis via an atypical polyketide synthase and P450-mediated cyclization. *Nature Communications* 9: 5281.

Huang JP, Fang C, Ma X, Wang L, Yang J, Luo J, Yan Y, Zhang Y, Huang SX. 2019. Tropane alkaloids biosynthesis involves an unusual type III polyketide synthase and non-enzymatic condensation. *Nature Communications* 10: 4036.

Ping Y, Li X, Xu B, Wei W, Kai G, Zhou Z, Xiao Y. 2019a. Building Microbial Hosts for Heterologous Production of *N*-Methylpyrrolinium. *ACS Synthetic Biology* 9: 5281.

Ping Y, Li X, You W, Li G, Yang M, Wei W, Zhou Z, Xiao Y. 2019b. De Novo Production of the Plant-Derived Tropine and Pseudotropine in Yeast. *ACS Synthetic Biology* 8: 1257–1262.

Qiu F, Yan Y, Zeng J, Huang JP, Zeng L, Zhong W, Lan X, Chen M, Huang SX, Liao Z. 2021. Biochemical and Metabolic Insights into Hyoscyamine Dehydrogenase. *ACS Catalysis* 11: 2912–2924.

Qiu F, Yang C, Yuan L, Xiang D, Lan X, Chen M, Liao Z. 2018. A Phenylpyruvic Acid Reductase Is Required for Biosynthesis of Tropane Alkaloids. *Organic Letters* 20: 7807–7810.

Qiu F, Zeng J, Wang J, Huang JP, Zhou W, Yang C, Lan X, Chen M, Huang SX, Kai G, et al. 2020. Functional genomics analysis reveals two novel genes required for littorine biosynthesis. *New Phytologist* 225: 1906–1914.

Srinivasan P, Smolke CD. 2020. Biosynthesis of medicinal tropane alkaloids in yeast. *Nature* 585: 614–619.

Srinivasan P, Smolke CD. 2021. Engineering cellular metabolite transport for biosynthesis of computationally predicted tropane alkaloid derivatives in yeast. *Proceedings of the National Academy of Sciences* 118.

Comment 17: Line 858: change to described.

Response 17: We have corrected these spelling mistakes in the revised manuscript (Line 861).

Comment 18: Line 870: respectively

Response 18: We have corrected this spelling mistake in the revised manuscript (Line 874).

Comment 19: Line 877: change to described.

Response 19: We have corrected this spelling mistake in the revised manuscript (Lines 881).

Comment 20: Line 914: Italicize the “N”

Response 20: We have italicized the “N” in the revised manuscript (Line 918).

Comment 21: Check the reference list for accuracy. The author is listed incorrectly for reference 11 and reference 12 is duplicated again in reference 111. There may be other errors I didn't catch.

Response 21: We have corrected the mistakes in the references after careful checking.

Comment 22: Supplemental Fig 1 legend line 5: “the” is duplicated

Response 22: We have deleted the duplicated “the” in the legend of supplementary Figure 1 in the revised manuscript.

Comment 23: Supplemental Figure 11 – Change title to “the characterization of AbMPOs”

Response 23: We have changed the title of supplementary figure 11 to “the characterization of AbMPOs”.

Comment 24: Supplemental Figures 44 and 45: change PPR to PPAR

Response 24: We have corrected it in the revised supplemental Figures 44 and 45.

Supplementary Fig. 44. The positions of mTAs biosynthetic genes on the chromosomes of *A. belladonna*. The corresponding IDs and chromosomal positions of mTAs biosynthetic genes were provided in Supplementary Table 25.

Supplementary Fig. 45. The positions of mTAs biosynthetic genes on the chromosomes of *D. stramonium*. The corresponding IDs and chromosomal positions of mTAs biosynthetic genes were provided in Supplementary Table 26.

Comment 25: Supplemental Figure 48: The legend refers to a dashed black box but the figure shows a red box. Please update the legend.

Response 25: We have updated the legend by changing the “dashed black box” to “red box”.

Thank you for your helpful comments, which have helped us to improve our MS substantially.

Reviewer #3 (Remarks to the Author):

The authors have adequately and exhaustively addressed the comments raised. A big thank you for also making all data available.

We appreciate your helpful comments that improved our MS substantially. Thank you very much.

Reviewers' Comments:

Reviewer #1:

Remarks to the Author:

In the newly revised version of their manuscript, Zhang and coworkers addressed the last concerns raised by my previous reviewing.

Reviewer #2:

Remarks to the Author:

All comments have been addressed by the authors

Reviewer #1 (Remarks to the Author):

In the newly revised version of their manuscript, Zhang and coworkers addressed the last concerns raised by my previous reviewing.

Reviewer #2 (Remarks to the Author):

All comments have been addressed by the authors